

# Anomalous holiday precipitation over southern China

Jiahui Zhang[1], Dao-Yi Gong[1], Rui Mao[1], Jing Yang[1], Ziyin Zhang[2], Yun Qian[3]

[1]State Key Laboratory of Earth Surface Process and Resource Ecology/Academy of Disaster Reduction and Emergency Management, Beijing Normal University, Beijing, 100875, China

[2]Environmental Meteorology Forecast Center of Beijing-Tianjin-Hebei, China Meteorological Administration, Beijing, 100081, China

[3]Pacific Northwest National Laboratory, Climate Physics Group, Washington, 99352, USA

*Correspondence to*: Dao-Yi Gong (gdy@bnu.edu.cn)

**Abstract:** The Chinese Spring Festival (CSF) is the most important festival in China. Officially, this holiday lasts
approximately one week. Based on the long-term station observations from 1979 to 2012, this manuscript reports that during the holidays, the precipitation over southern China (108°E-123°E and 21°N-33°N, 155 stations) has been significantly reduced. The precipitation frequency anomalies from the fourth day to the sixth day after Lunar New Year's Day (i.e., days [+4, +6]) were found to decrease by -7.4%. At the same time, the daily precipitation amounts experienced a reduction of -0.62 mm d$^{-1}$ during days +2 to +5. The holiday precipitation anomalies are strongly linked to the relative humidity (ΔRH)
and cloud cover. The station observations of the ΔRH showed an evident decrease from day +2 to +7, and a minimum appeared on days [+4, +6], with a mean of -3.9%. The ΔRH vertical profile displays a significant drying below approximately 800 hPa. Between 800 hPa and 1000 hPa, the mean ΔRH is -3.9%. The observed station daytime low cloud cover (LCC) evidently decreased by -6.1% during days [+4, +6]. Meanwhile, the ERA-Interim daily LCC also shows a comparable reduction of -5.0%. The anomalous relative humidity is mainly caused by the lower water vapor in the lower-
middle troposphere. Evident negative specific humidity anomalies persist from day -3 to day +7 in the station observations. The average specific humidity anomaly for days [+4, +6] is -0.73 g kg$^{-1}$. When the precipitation days exclude the mean, the anomaly remains significant, being -0.46 g kg$^{-1}$. A significant deficit of water vapor is observed in the lower troposphere below 700 hPa. Between 800 hPa and 1000 hPa, the mean specific humidity dropped by -0.70 g kg$^{-1}$. This drier lower-middle troposphere is due to anomalous northerly winds. Authors have proposed that the anomalous atmospheric circulation
is likely related to the holiday aerosol anomaly. Station and satellite observations show that the East Asian aerosol concentrations during the CSF decrease evidently, the largest reduction occurring on days [-3, -1]. At the same time, a concurrent cooling is observed in the lower troposphere. In addition, an anomalous low pressure tilting westward occurs in the troposphere over East Asia. The anomalous cold advection seems to help trigger/strengthen a cyclonic circulation anomaly, which is responsible for the northerly winds and the less precipitation around the holidays. This possible
mechanism needs further clarification by elaborate observation analysis and modeling.



# 1 Introduction

The role aerosols play in weather/climate variability is among the most important. Particularly the large uncertainty involving the clouds, precipitation, and radiation makes it difficult to accurately estimate their contributions to the long-term climate changes (Boucher et al., 2013). Recent studies have noted that short-term air pollution associated with intense economical/political/cultural events has discernable influences on local and/or regional weather worldwide, where air pollution caused by regular or occasional anthropogenic emissions significantly impacts atmospheric physics and chemistry (e.g., Zhao et al., 2006; Sanchez-Lorenzo et al., 2012; Williams et al., 2015). Recent estimation indicates that in the fast-developing China, anthropogenic aerosols might have contributed 10% ±4% of the current global radiative forcing (Li et al., 2016a). However, the possible mechanism involving the cloud-precipitation processes and interactions over monsoonal Asia remains among the most challenging topics (Li et al., 2016b).

Alternating weekdays and weekends may be the most apparent regular activities of human beings. The so-called weekly cycle effects of aerosol concentrations and atmospheric parameters, which are strongly related to the aforementioned regular activity, have been widely reported globally (Gordon, 1994; Cerveny and Balling, 1998; Bäumer et al., 2008). At first, these studies of the weekly cycles were focused on cities and industrialized areas. Later, it was reported that the weekly cycles of air pollution and atmospheric variables are not only local phenomena confined within urban areas but are also large-scale phenomena that are closely related to anthropogenic aerosol effects (Gordon, 1994). Over developed Europe, for example, Bäumer et al. (2008) analyzed the station data from Aerosol Robotic Network over central Europe and reported that the aerosol optical thickness is higher midweek than on the weekend. In the rapidly industrializing Asia, there are also discernible weekly cycles. For example, Gong et al. (2007) found that the $PM_{10}$ (particulate matter with a diameter of less than 10 μm) values in China experience significant weekly cycles with ranges of approximately 2 μg m$^{-3}$, where a peak appears on Thursday and a low appears on the weekend. Accompanying the weekly cycles of $PM_{10}$, there are robust co-variations in the surface solar radiation, total cloud cover, maximum temperature, and relative humidity. During the weekdays, the total cloud cover and relative humidity are lower, while the surface solar radiation and maximum temperature are higher. In contrast, the higher cloud cover and humidity and the lower radiation and maximum temperature appear on the weekends. Evident weekly cycles are also identified in other parameters, such as the diurnal temperature range (Gong et al., 2006) and thunderstorm frequency (Yang et al., 2016). In South Korea, Choi et al. (2008) found that the summertime cloudiness shows a bell-shaped curve between midweek and weekend. Meanwhile, the aerosol concentrations are higher midweek than on the weekend. The mechanisms of these weekly cycles are not yet fully understood. Gong et al. (2006) supposed that the direct or indirect aerosol effects may play different roles depending on the different climatological circumstances. Gong et al. (2007) suggested that during the boreal summer, the weekly cycles in eastern China are likely associated with aerosol-atmosphere interactions, where the accumulated aerosol concentrations play a key role in triggering the atmospheric instability on Thursday. Wang et al. (2012) emphasized that during the autumn, the semi-direct aerosol effects likely play a dominant role in the weekly cycle of cloud cover in southeastern China. Yang et al. (2016) proposed the



importance of the aerosol types. In central China, aerosol absorption is strong, which suppresses convection more on weekdays, whereas in southeast China, aerosols are more hygroscopic, which helps invigorate thunderstorms on weekdays.

In addition, there are occasional anthropogenic air pollution events (in clean as well as the dirty cases). For example, Xin et al. (2012) found that the concentrations of $PM_{2.5}$ in Beijing-Tianjin-Hebei area were reduced by approximately 50%

with the implementation of emission controls by the government during the Beijing Olympic Games in the summer of 2008. Similar actions were also implemented when China hosted the Asia-Pacific Economic Cooperation meeting in 2014 and held the national military parade in 2015; as a result, the air quality was considerably improved over northern China (Chen et al., 2015; Wang et al., 2015; Xu et al., 2015; Wen et al., 2016; Xu et al., 2017). In contrast, setting off fireworks during holiday celebrations can produce severe air pollution across the world; for example, on the Fourth of July holiday in America

(Carranza et al., 2001), the New Year's fireworks in Mainz (Drewnick et al., 2006), during the Diwali festival in India (Singh et al., 2014), during the Chinese Spring Festival (CSF) (Tang et al., 2016), etc. The emission and pollutant components may differ among these events (Wang et al., 2007; Zhang et al., 2010; Tang et al., 2016). Note that the CSF holiday repeatedly occurs every year in January-February across the whole country and is the most important holiday in China. Gong et al. (2014) indicated that during the CSF holidays, the $PM_{10}$ concentrations were reduced by approximately

9.24% due to the economic slowdown. Lin and McElroy (2011) also estimated that during the celebration of the CSF there is a notable reduction in anthropogenic emissions. According to their estimations for 2005, 2007, 2008, and 2010, the CSF contributes to an $NO_x$ emission reduction of 12%. In addition, the 2009 CSF contributes to a reduction of 10%. Some studies have tried to identify the possible consequences for atmospheric physics and weather anomalies resulting from these local and regional occasional pollution events. As an example, in the United States, the diurnal temperature range increased

significantly during the three days after 11 September 2001 when civil aviation planes were grounded (Travis et al., 2002). Gong et al. (2014) analyzed the temperature anomalies during the CSF holidays during 2001-2012 and found that there are significant negative anomalies of approximately -0.81°C in eastern China, wherein atmospheric feedbacks likely play important roles in enhancing the cooling.

The purpose of the present study is to investigate the possible changes of precipitation over China during the CSF

holidays. The paper is organized as follows. The data and method are presented in Section 2. Results, including the anomalous precipitation frequency and amount, the relevant changes in humidity, cloud cover and temperature, and the anomalous atmospheric water vapor and circulation, are described in Section 3. The possible relevance of these results to the holiday aerosols is discussed in Section 4. In addition, Section 5 presents a summary.

## 2 Data and methods

### 2.1 Data and study area

The daily precipitation data analyzed in the present study were obtained from the China Meteorological Administration observation archives. The daily precipitation amounts as well as the precipitation types are recorded. There are six kinds of





readings, namely, (a) no precipitation; (b) trace precipitation (daily precipitation readings of less than the measurable 0.1 mm); (c) precipitation from fog, frost and dew; (d) precipitation from pure snow; (e) precipitation from snow and rain; and (f) precipitation from rainfall. Because the frequency of snowfall events is low in the warm area of southern China and the precipitation from fog and dew is considerably small (accounting for 0.14% of our data), in this study, we considered

only the precipitation amounts and ignored the precipitation types. This dataset consists of 917 stations, and the earliest observations start in the 1950s. Here, we confine our analysis period to 1979-2012, during which China experienced a rapid development and massive emissions of air pollutants. Since we aim to investigate the precipitation around the Lunar New Year's Day, in this study, only the daily precipitation from January to February is employed. After excluding the 319 stations with missing values, we retained the 598 stations without missing data during 1979-2012.

Prior to this analysis, we have to determine the target area based on the precipitation climatology. Because the Chinese Lunar New Year occurs in January or February, here, we investigated the climatological means of the precipitation amounts as well as the precipitation frequencies based on the observations from 598 stations. As shown in Figure 1a, in the boreal winter, the number of precipitation days is much greater in southern China than in the north. A maximum of >35 days occurs between approximately 25-30°N and 102-115°E, while the precipitation frequency is less than 15 days north of 35°N. At the

same time, the major precipitation belt, with precipitation amounts > 50 mm, also appears over the southeast of mainland China, being generally consistent with the spatial features of the frequency patterns (Figure 1a). Although the number of precipitation days in the Sichuan Basin is high, the precipitation amounts are not such large. In addition, Sichuan Basin has a unique climate due to its special basin topography. Therefore, we did not include it in our study area. Finally, we confined our study area to 108°E-123°E and 21°N-33°N (as indicated by the black dashed rectangle in Figure 1). Within this area,

there are 155 available meteorological stations. Among these stations, the minimum number of precipitation day (15.7 days) appeared at the Zhumadian station (33°N, 114°E), and the minimum precipitation amount (15.3 mm) appeared at the Ankang station (32°N, 109°E). Meanwhile, the maximum precipitation days (41.6 days) and maximum precipitation amount (214.5 mm) were found at Wugang (26°N, 110°E) and Nanyue (27°N, 112°E), respectively. On average, the precipitation frequencies and amounts during the period of 1979-2012 are 30.3 days and 124 mm. In addition to the daily precipitation, we

also analyzed other meteorological variables, including the mean temperature and relative humidity.

The atmospheric circulation and moisture in the troposphere were taken from the ERA-Interim reanalysis datasets (Dee et al., 2011), which have a 1.5 ° spatial resolution and span the period of 1979-2012. We utilized surface parameters, such as the cloud cover and total column water vapor. In addition, the impacts of the air temperature, relative humidity, and horizontal winds on pressure levels were also investigated. Only the data at step 0 for 00:00, 06:00, 12:00, and 18:00 UTC

were obtained and subjected to analysis. Averaging these 4 times yields the daily mean.

**2.2 Analysis method**

We employed a temporal composite analysis to investigate the possible changes in the daily precipitation and the statistical significance. This was carried out via the following steps. First, we identified all Lunar New Year's Days (LNYD) from





1979 to 2012. As listed in Table 1, the New Year's Days all fall within late January to mid-February. During our analysis period, the earliest New Year's Day is on 22 January, and the latest occurs on 20 February. These LNYDs are denoted as day 0. The days before and the day after the LNYD are denoted as day -1 and day +1, respectively. Accordingly, two days before and after the LNYD are denoted as day -2 and day +2, etc.

Second, we estimated the anomalies from the climate means. When dealing with continuous variables (such as the temperature, humidity, atmospheric circulation, and water vapor), we calculated the long-term means for each calendar day from 1 January to 28 February. Then, the means were subtracted from the daily observations to determine the corresponding anomalies. This process is same as that used in Gong et al. (2014). In contrast to these continuous variables, precipitation occurs quite randomly. For a specific day, say day 0, we counted the number of precipitation days ($N$) during the whole data

period from 1979-2012. Thus, the precipitation frequency (hereafter denoted to as $F$) is deduced by:

$$F = \frac{N}{34} \times 100\% ,$$                            (1)

     The climatic means of $F$ were calculated based on Gregorian calendar days. Taking day 0 as an example, after reading all 34 corresponding Gregorian calendar dates we computed their mean precipitation frequencies. Their averages represent the climatic means. Note that when computing the climatic means, we used a 7-day window because the typical synoptic

time scale is approximately 7 days. We expect this process to serve as a filter to suppress the random synoptic noise. The departure of $F$ from the climatic mean is the anomalous frequency ($\Delta F$). To determine a more robust signal, we used a 3-day window. In other words, when computing the $F$ for day 0. We used all data from day -1 to day +1.

## 3 Results

### 3.1 Significant reduction of precipitation

We computed the daily anomalies of the precipitation frequencies for each of the stations. To determine how the precipitation might change before and after New Year's Day, we considered days from -12 to +12. This method is similar to that of the work of Gong et al. (2014). We also tried to use a longer time window and found that the conclusions remain the same. To focus on southern China, we computed the means for all 155 stations within the target area (Figure 1). The regional mean $\Delta F$ is shown in Figure 2a. Although the $\Delta F$ before the LNYD is generally positive, the salient feature is the $\Delta F$

reduction after the LNYD. The mean for day +3 to +7 is -5.5%, and a minimum of -9.4% occurs on day +5. Note that the largest reductions occur on days [+4, +6], where the averaged anomalies are as low as -7.4%. Precipitation is a natural phenomenon. Thus, its frequency anomaly should be spatiotemporally random for a certain day from 1979-2012 among all the stations over this large region. The standard deviation of the stations' $\Delta F$ would provide information concerning further uncertainties. A smaller standard deviation suggests that similar anomalies tend to be observed, thus indicating a robust

signal, whereas a larger standard deviation implies diverse changes. We found that the standard deviation of the days [+3, +7]



is approximately ±3.7%, being smaller than most days before the LNYD. Note that the means may be biased by outliers or skewness. Thus, to determine the diverseness of these frequency anomalies among the 155 stations, we also computed the medians and the lower and upper percentiles. In Figure 2a, the ranges of the 10th and 90th percentiles are plotted as the error bars and are shown with the medians. The 90th percentiles for the days [+4, +6] are clearly -0.9%, -3.9%, and -1.7% respectively, all being significantly below 0. At the same time, the medians are generally similar to the means; for example, on days [+3, +7], the means (-3.1%, -6.5%, -9.4%, -6.3%, and -2.2%) and medians (-3.6%, -7.0%, -9.8%, -7.0%, and -1.8%) are almost identical, likely suggesting that the majority of the anomalies tend to be negative and that the means are not skewed by large departures or outliers.

We further investigated the statistical significances of the precipitation anomalies by employing a Monte-Carlo approach (e.g., Stjern, 2011; Wang et al., 2012). Here, the Monte-Carlo test was performed by randomly rearranging the sequences of the lunar calendar days. For each experiment, we first generated the random sequence and then estimated the frequencies for a specific day station by station. In addition, the regional anomaly mean was calculated as described above. We repeated these steps 1000 times and obtained the 10th and 90th percentiles of these experiments. By comparing observations with these Monte-Carlo-yielded percentiles, we might estimate whether the frequency anomalies are beyond those of random chance. If the original observations are lower than the 10th percentile, these measurements are significant at the 0.1 level.

For the regional mean frequency anomalies, there are few cases where the simulated anomaly is smaller than the observations during the holidays. Only 1 of 1000 random Monte-Carlo experiments is below the observation on day +5. On days +4 and +6, there are 10 and 14 cases with values smaller than the observation. On days +3 and +7, there are slightly more (135 and 190, respectively). We found that the 10th percentiles during days +4 to +6 are -3.6%, -3.4% and -3.5%, respectively. Note that the observations (-6.5%, -9.4% and -6.3%) are all below the corresponding 10th percentiles of the Monte-Carlo experiments. Therefore, we may conclude that the observed $\Delta F$ is significant at the 0.1 level from days +4 to +6. We also examined the medians of the 155 stations and found a similar result. From day +4 to +6, there are only 9, 1 and 7 cases with values below the observations. The observed medians during days +4 to +6 are -7.0%, -9.8% and -7.0%, which are all below the Monte-Carlo 10th percentiles (-3.7%, -3.6% and -3.6%, respectively).

We further investigated the possible changes of daily precipitation amounts. Here, all trace precipitation records are excluded. The composite method is similar to that for frequency. In other words, for a specific lunar calendar day, the anomalous amount is calculated as the mean daily precipitation amount minus the climate reference, where the climate reference is estimated based on the Gregorian calendar. Then, we averaged the anomalies for all stations to obtain a regional mean. Figure 2b shows the result. Unlike the precipitation frequency, the amount shows no evident departures before the LNYD. Interestingly, from day +2 to day +5, the amounts experience continuously negative anomalies with a mean of -0.62 mm d$^{-1}$. In particular, a minimum of -1.0 mm d$^{-1}$ occurs on day +3. During days +2, +4, and +5 the anomalies are similar (being -0.54, 0.52, and -0.48 mm d$^{-1}$, respectively). Differing from the precipitation frequency, the amount ranges from the 10th-90th percentiles are obviously larger. In addition, during days [+2, +5], the upper bounds all exceed zero and



imply larger uncertainties. Thus, the gradual decrease in the amount after the LNYD is likely consistent with the significant reduction in precipitation frequency. Note that the Monte-Carlo test suggests a high confidence of the negative amount anomalies during the holidays. Of 1000 Monte-Carlo experiments, there are 83, 14, 75 and 70 cases with values smaller than the observations during days [+2, +5]. The Monte-Carlo $10^{th}$ percentiles for days [+2, +5] are -0.47, -0.45, -0.42 and -

0.40 mm $d^{-1}$, respectively, whereas the corresponding observations of -0.54, -0.95, -0.52 and -0.48 mm $d^{-1}$ are all smaller. We also examined the median of the 155 stations by employing the Monte-Carlo test. The result is similar. There are 70, 16, 66 and 43 experimental values that are smaller than those observed. The medians of the observation for these days (-0.63, -0.97, -0.65 and -0.70 mm $d^{-1}$, respectively) are all smaller than the corresponding Monte-Carlo $10^{th}$ percentiles (-0.56, -0.54, -0.52 and -0.47, respectively). Therefore, the means and medians of the amount anomalies during days [+2, +5] are

significant at the 0.1 level. Note that the large error bars in Figure 2b may imply somewhat larger diversities/differences of the daily precipitation amounts among the 155 stations. The Monte-Carlo test likely provides a more reasonable estimation of the significance. With the above analysis, we may conclude that there is a significant reduction in the precipitation frequency and a decrease in the daily precipitation amount after the LNYD.

## 3.2 Spatial distribution of the holiday precipitation anomalies

Although the precipitation reduction after the LNYD is evident in the composite analysis based on the regional means (Figure 2), this feature might differ among stations. In this section, we investigate the spatial distribution of the precipitation anomalies. A significant frequency reduction occurs on days [+4, +6], and a significant amount reduction appears on days [+2, +5]. Here, we computed the mean frequency anomalies during days [+4, +6] and the mean amount anomalies during [+2, +5] for each station. The statistical significance is also estimated using the Monte-Carlo test at each station. The method used

is similar to that described in Section 3.1. Here, the Monte-Carlo test was performed for every station by randomly rearranging the lunar calendar, and the simulation was repeated 1000 times. The $10^{th}$ ($90^{th}$) and $20^{th}$ ($80^{th}$) percentiles were taken as the criteria. For the precipitation frequency, the mean value is regarded as significant at the 0.1 (or the 0.2) level only when the anomalies in all three days (i.e., days +4, +5 and +6) were significant at the 0.1 (or 0.2) level. Similarly, when the daily precipitation amounts on all four individual days (from day +2 to +5) are significant, their mean is regarded as a

significant anomaly. The results are shown in Figure 3.

Figure 3a shows that the majority of the $\Delta F$ have negative signs. Among the 155 stations, only 7 stations have positive signs, and none of these positive anomalies are significant. Of the 148 stations with $\Delta F$ reductions, 57 stations are significant at the 0.1 level, and 108 are significant at the 0.2 level. The $\Delta F$ reduction center is mainly located over the central region (approximately 108°E-115°E and 28°N-32°N). The $\Delta F$ minimum reaches -13.5% at the Sansui station (27°N, 108°E).

Looking at the anomaly's magnitude, there are 55 stations with $\Delta F$ reductions exceeding -9%. Among these 55 stations, the anomalies for 45 stations are significant at the 0.1 level, and those at the other 10 stations are significant at the 0.2 level.





Unlike the frequency, the spatial features of the amount anomalies differ among stations (Figure 3b), which is consistent with the relatively larger span of the 10th-90th percentiles, as shown in Figure 2b. Nevertheless, there are still 117 stations with decreased amounts during days [+2, +5]. Among these, 27 stations showed decreases significant at the 0.2 level, and 8 stations showed those significant at the 0.1 level. The largest reduction is -2.41 mm d$^{-1}$, which appeared at the

Shouxian station (32 °N, 116 °E).

Based on the above analysis of the temporal composites and spatial distributions, we found that the significant reductions of precipitation frequency and daily precipitation amount occur regionally within an approximate one-week period after the LNYD.

## 3.3 Relative humidity and cloud anomalies

Relative humidity (RH) is the most important factors that directly controls/influences precipitation. In this subsection, we investigate the possible changes in the relative humidities associated with the holiday precipitation decreases. Here, we examined the station data to analyze the relative humidity by making composites. Note that the method of estimating the anomalies is different from that for precipitation frequencies. The composites of the relative humidity (as well as those of other meteorological parameters analyzed in the following sections) are based on anomalies (ΔRH), which are obtained by

subtracting the 1979-2012 long-term mean from the daily relative humidity values.

For each lunar calendar day, all 34 daily anomalies are collected and subjected to a *t*-test with the null hypothesis of their mean being not significantly different from zero at the 0.05 (or 0.1) level. In Figure 4, we plotted the means during days [-12, +12]. To facilitate a comparison, the range of one standard error estimated from the 34 anomalies is shown via error bars. The ΔRH experiences an evident decrease from day +2 to +7, with a mean value of -2.5%. The driest value is -4.6% at

day +5. The mean ΔRH of the lowest anomalies for days [+4, +6] is -3.9%, which is significant at the 0.1 level as estimated from a left-tail *t*-test. Note that the drying (wetting) may be the result of less (more) precipitation, as demonstrated in the previous sections. For clarity, we also computed the ΔRH using only the data from no-rain days. We found that the ΔRH for no-rain cases shows a similar temporal feature (figure not shown). The lowest values still occur on days [+4, +6], with anomalies of -0.97%, -0.69% and -0.38%, respectively. However, these anomalies are not statistically significant. This result

could be caused by the relatively smaller data sample and excluding the precipitation days. In any case, the similar drying features on no-rain days may provide support for the theory that the lower ΔRH could be a cause of the precipitation reduction and may even help enhance the drying.

We also examined the spatial distribution of the ΔRH. The station means of the ΔRH during days [+4, +6] are plotted in Figure 5. All 155 stations show negative anomalies, with a maximum and a minimum of -0.6% and -8.1%, respectively.

When estimating the statistical significance, we consider the mean for days [+4, +6]. Thus, we did not consider the significance of each of the three days. If the resulting anomaly is significant at the 0.05 (or 0.1) level as calculated by a *t*-test, this station is regarded as significant during days [+4, +6]. As shown in Figure 5, there are 70 (98) stations that are





significant at the 0.05 (0.1) level. Additionally, we investigated the spatial features of no-rain days and found that most ΔRH during days [+4, +6] were also negative (the number of days is 136, accounting for approximately 88% of the total stations), and the lowest anomaly is -7.6% (figure not shown). Again, this result suggests a drying atmosphere near the surface during the New Year's holiday.

For the stratification of precipitation during the winter season, the relative humidity in the middle-lower troposphere is more important than the surface humidity. The precipitation reduction could occur with a drier higher atmosphere. We then investigated the ΔRH in the low-middle level troposphere using the ERA-Interim reanalysis datasets of the pressure levels from 1979 to 2012. Here, we selected 99 grid points between 21 °N-33 °N and 108 °E-123 °E. We examined the ΔRH values at each level from 1000 hPa to 500 hPa. Generally, the results are similar, showing drying tendencies throughout, as expected.

Note that the ΔRH values in the low levels are more evident. At 1000 hPa, the ΔRH has a continuous reduction during days [+3, +6], with a mean decrease of -3.8%. The negative anomalies on these days are significant at the 0.1 level. The largest reduction occurs on day +5, being -5.2% (figure not shown). In Figure 4, we plot the regional mean ΔRH at 850 hPa. A similar reduction is observed during days [+2, +6], where the lowest anomalies are still found on days [+4, +6]. In addition, the values on days +4 and +5 (-4.0% and -5.7%) are both significant at the 0.1 level. In the middle-upper troposphere, the

ΔRH during the holidays is not evident.

      The spatial distributions of ΔRH at 1000 and 850 hPa during days [+4, +6] were also analyzed. At 1000 hPa, the majority of the significantly negative ΔRH values were located over southern China and the neighboring western Pacific, with a northeast-southwest distribution. The results at 850 hPa are plotted in Figure 5, and only the significant values (above the 0.1 level) are shown. At 850 hPa, the significant region is smaller spatially but has a greater magnitude than that at 1000

hPa. Over the land, the majority of the anomalies are between -4% and -6% at 1000 hPa, whereas the values are between -6% and -10% at 850 hPa. The minimum grid point value at 1000 hPa is -6.5%, while the minimum is as low as -9.7% at 850 hPa.

      The drying in the lower troposphere can be more clearly seen in the relative humidity profiles. The vertical profile of the ΔRH during days +4 to +6, as averaged over all 99 grids in southern China, is shown in Figure 6. Because the meaningful changes occurred in the lower troposphere, here we plotted only the ΔRH values below the 500 hPa level. Below 500 hPa,

the anomalies all have negative signs. In addition, the significant ΔRH values appear below approximately the 800 hPa level (Figure 6a). The negative anomaly at 850 hPa is -3.9%, and that at 1000 hPa is -4.2%. The mean for these layers (below 800 hPa) is -3.9%. Generally, the significant drying of the relative humidity appears in the low-middle level, which physically agrees with the precipitation reduction.

      As the relative humidity at the low-middle level decreases, consistent changes in clouds should occur, particularly in the

low-level cloud coverage (LCC). To confirm this, we first analyzed the surface-observed cloud data, which are from the Global Surface Weather Dataset obtained from the China Meteorological Administration. In this dataset, the observations are taken 4 times per day (00:00, 06:00, 12:00, 18:00 UTC) during the years 1980-2012, except for during 2000, which is not available. Only those days with all 4 records are included. Of the 304 stations in our study area, we selected 137 stations





with overall data availabilities above 30%. When deriving the anomalies, we applied the same method as that used for ΔRH. The results show that the total cloud cover showed no evident changes, whereas the LCC experiences a significant decrease during the New Year's holiday. The mean anomaly during days [+1, +5] is -2.6%. In addition, a minimum of -3.1% appears on day +5, which is significant at the 0.1 level. We should note that sometimes the LCC cannot be completely distinguished

from the medium cloud cover in the station observations. Moreover, the cloud cover changes rapidly and might depend on the observer. At nighttime, the uncertainty of the observations is much greater. We would expect more reliable observations during the daytime. The mean LCC of 00:00 and 06:00 UTC is computed and regarded as the daytime observation. As shown in Figure 4b, the daytime LCC shows an evident reduction during days [+4, +6]. The values of each of three days are all significant at the 0.1 level, as estimated using a left-tail $t$-test. The mean for the three days is -6.1%, and a minimum of -

9.7% occurs on day +5. Compared to the climate daytime cloud cover of 78% during January to March in southern China, the LCC reduction during the New Year's holiday has a considerably large magnitude. Figure 5b shows the spatial distribution of the daytime LCC values during days [+4, +6]. For a total of 137 stations, 119 (86.9%) stations showed negative signs. In addition, there are 8 (27) stations with values significant at the 0.05 (0.1) level, as estimated via a both-tail $t$-test. A minimum of -16.8% occurs at the Fuyang station (33 °N, 115 °E).

To assess the robustness of the cloud cover changes, the cloud cover of the ERA-Interim data was investigated. We found that the high cloud cover and the medium cloud cover showed no significant changes (figure not shown). Interestingly, there is a significant reduction in the LCC during the New Year's holidays. Figure 4d shows the temporal variations of the ERA-Interim LCC. The LCC clearly shows an outstanding reduction during days [+4, +6]. The largest reduction, i.e., -5.9%, appears on day +5. The values from day +4 to day +5 are all significant at the 0.1 level, as calculated by a left-tail $t$-test.

Averaging over days [+4, +6], the mean value is -5.0%. This magnitude is comparable to the station-based daytime LCC anomaly (-6.1%). The spatial distribution of the LCC anomalies for the ERA-Interim data during days [+4, +6] is shown in Figure 5d. The LCC reduction center is located in the eastern region and has a magnitude between approximately -8% and -12%. This reduction is almost identical to that of the station observation (c.f., Figure 5b, d). Note that in the ERA-Interim data, the cloud height is defined according to the corresponding sigma level. The low clouds correspond to sigma 0.8-1.0. A

significant decrease of the relative humidity occurs below 800 hPa (Figure 6a), which physically agrees with the negative LCC anomalies.

### 3.4 Temperature and water vapor anomalies

The negative ΔRH during the holidays might be caused by anomalous temperatures, by the water vapor or both, depending on certain conditions. In this subsection, we analyze the changes of the temperature and water vapor associated with the

holiday precipitation reduction. First, we examined the temporal changes of the temperature over southern China using the same composite analysis as that used for the relative humidity. We found that there is a continuous cooling of the daily mean temperatures from day -3 to day +6, which are all significant at the 0.1 level, as estimated by a left-tail $t$-test (figure not shown). The mean temperature anomaly of these days is -1.12°C. The maximum and minimum temperature anomalies are -





0.70°C and -1.42°C for days +6 and +3, respectively. Precipitation often causes a low temperature. To exclude the possible influence of precipitation on temperature, we also examined the daily temperatures for no-rain days and found a similar temperature anomaly of -1.22°C for days +1 to +6. In addition, there is a similar minimum of -1.48°C. These values for the no-rain days are significant at the 0.05 level.

Note that Gong et al. (2014) have reported negative temperature anomalies over the whole of eastern China around the LNYD when analyzing the shorter data period of 2001-2012. They found that the most significant decrease appears on days -3 to +2, with a mean value of -0.81°C. The temperature beyond this period over the southern China, however, was not addressed in their analysis. As demonstrated in the previous sections, the most significant reductions of the $\Delta F$ and $\Delta RH$ are observed on days [+4, +6]. Thus, we further computed the mean of the temperature anomalies ($\Delta T$) for days [+4, +6] and

estimated their statistical significances using a *t*-test for each of the stations. The results show that all 155 stations have negative anomalies (figure not shown). The maximum $\Delta T$ of -0.28°C occurs at the Fengjie station (31°N, 109°E), and a minimum of -2.14°C appears at the Jiuxian station (25°N, 118°E). The anomalies for 116 stations are statistically significant at the 0.1 level, and those for 95 stations are significant at the 0.05 level. We also investigated the temperature anomalies during days [+4, +6] using no-rain days and found similar results. All stations continue to show negative anomalies, and

more stations are significant (134 stations exceed the 0.1 level, 117 stations exceed the 0.05 level).

    The above spatial distribution suggests that temperature anomalies are unlikely at the local scale. For clarity, we also investigated the vertical profiles and the spatial distributions of $\Delta T$ using the ERA-Interim pressure level data. The mean $\Delta T$ values below the 500 hPa level during days [+4, +6] are averaged for all 99 grid points over southern China and plotted Figure 6b. The significant negative temperature anomalies below 700 hPa are evident, where all values are as low as < -1°C.

The mean for these layers is -1.37°C. To determine whether the cooling of lower-middle troposphere is a regional-scale phenomenon, we further analyzed the $\Delta T$'s spatial distribution. Note that the minimum of -1.56°C occurs at the 850 hPa level in Figure 6b. Here, we computed the 850 hPa $\Delta T$ during days [+4,+6] for each of the grid points (figure not shown), revealing that the majority of the study area experiences a significant cooling (exceeding -1°C) that is almost identical to that from the surface station observations. Based on these analyses, we may conclude that during the holidays (particularly, days

[+4, +6]), there is an anomalous temperature cooling over southern China from the surface to the middle troposphere (below 500 hPa).

    Cooling favors condensation and precipitation. If the water vapor content remains a constant, a cooler temperature should result in a higher $\Delta RH$. During the period of 1979-2012, over southern China, the climatic mean temperatures for days [+4, +6] at 1000 hPa and 850 hPa are 9.3°C and 3.3°C, respectively. Meanwhile, the specific humidities at 1000 hPa

and 850 hPa are 5.5 g kg$^{-1}$ and 4.4 g kg$^{-1}$. According to the Clausius-Clapeyron equation (Murray, 1967), the climatological relative humidities are, respectively, 75.4% at 1000 hPa and 77.5% at 850 hPa. The observed coolings of -1.15°C at 1000 hPa and -1.56°C at 850 hPa (Figure 6b) could cause corresponding $\Delta RH$ increases of +6.0% and +9.1%, respectively. This



contradicts the observed negative ΔRH anomalies (c.f., Figure 5). Therefore, the cooling temperature is not a direct factor causing the drier ΔRH and the precipitation reduction.

Alternatively, the water vapor should be responsible for the anomalous ΔRH. We analyzed the station specific humidity (SH). Here, the SH is estimated from the surface pressure, relative humidity and temperature, where the Tetens formula is

employed to estimate the saturated vapor pressure over water (Murray, 1967). Figure 7a shows the specific humidity anomalies (ΔSH) before and after the LNYD. Evident negative anomalies persist from days -3 to +7. The lowest values appear at approximately days [+4, +5]. Note that the negative ΔSHs during days +2 to +6 are significant at the 0.05 level, with a mean of -0.71 g kg$^{-1}$. A minimum of -0.82 g kg$^{-1}$ occurs on day +4. In addition, the average ΔSH for days [+4, +6] is -0.73 g kg$^{-1}$. We repeated the composite analyses for no-rain days and found that the ΔSH have similar continuous reductions

after New Year's Day (Figure 7b). The ΔSHs of days [+2, +5] are significant at the 0.1 level, and their mean is -0.43 g kg$^{-1}$. The minimum of -0.54 g kg$^{-1}$ also appears on day +4, while the mean from day +4 to +6 is -0.46 g kg$^{-1}$, being smaller than the values estimated for all days. At the same time, the spatial distribution of ΔSH during days [+4, +6] reveals a regional-scale reduction over the study area (Figure 8). All stations clearly show negative anomalies, both when all days are analyzed and when only no-rain days are analyzed. In the former instance, the ΔSH values for the 140 stations is statistically

significant at the 0.05 level, and in the latter case, the number of stations with statistically significant values is 102.

We further investigated the ΔSH values in the low-middle troposphere using the ERA-Interim reanalysis data. The ΔSH at 1000 hPa (850 hPa) during days [+2, +6] are all significant at the 0.05 (0.1) level, with a mean of -0.75 (-0.65) g kg$^{-1}$. It is clear that the ΔSH at 1000 hPa and 850 hPa display features similar to those of the surface observations in both their temporal variations and the magnitudes of the negative anomalies during the holidays (c.f., Figure 7). Furthermore, we

investigated the spatial distributions of ΔSH on days [+4, +6]. The evident ΔSH reduction covers almost the whole of southern China, with minimums extending from southeastern China to the western North Pacific, south of approximately 30°N (Figure 8). The anomaly center lies between 110°E-130°E and 20°N-30°N.

The drying of the ΔSH in the low-middle troposphere is more obvious, as seen in the vertical profile (Figure 6c). The significant negative ΔSH values appear in the levels below 700 hPa. Below 800 hPa, the ΔSH values are all below -0.50 g

kg$^{-1}$. The mean for these layers (800 hPa to 1000 hPa) is -0.70 g kg$^{-1}$. Similarly, we also estimated the variations of the relative humidity corresponding to these ΔSHs. The ΔSHs at 1000 hPa and 850 hPa are -0.78 g kg$^{-1}$ and -0.69 g kg$^{-1}$, respectively (Figure 6c). If the temperature remains unchanged, these values would reduce the relative humidity by -10.9% and -12.3%, respectively. Obviously, the reduction of the water vapor in the lower-middle troposphere plays the dominant role in causing anomalous relative humidities, low-level clouds and precipitations.

The total column water vapor (TCWV), or the precipitable water, is also a large-scale factor that correlates well with precipitation (e.g., Qian et al., 2009). The drying from the surface to the mid-troposphere is suggestive of the reduction in the TCWV. For clarification, we performed a composite analysis of the ERA-Interim TCWV. Unsurprisingly, the ΔTCWV experiences a continuous reduction from day +1 to day +5 (Figure 9a). The ΔTCWV are significant at the 0.05 level during




days [+2, +5], with a mean of -1.93 kg m$^{-2}$. In addition, the largest negative ΔTCWV values are observed on days +4 and +5 (2.43 kg m$^{-2}$ and 2.36 kg m$^{-2}$, respectively). The spatial distribution of the ΔTCWV values during days [+4, +6] is shown in Figure 9b. The TCWV decreases by approximately -1.00 kg m$^{-2}$ or more over most of the southern China. The minimum is located over the neighboring western North Pacific, between 120°E-130°E and 20°N-30°N. Meanwhile, positive ΔTCWVs

with somewhat smaller magnitudes appear to the east (over approximately 150°E-175°E and 20°N-30°N). Generally, the reduction in the TCWV over the western North Pacific and eastern China is physically consistent with the negative anomalies of the SH over southern China. The observed decrease in precipitable water during the holidays is unfavorable for precipitation.

Based on the above analysis, we may conclude that, although the relative humidity (as well as the relevant cloud cover

and precipitation) would benefit from the cooling temperatures, its reduction during days [+4, +6] is strongly dominated by the drying water vapor in the lower troposphere. Therefore, the reduction of the SH is likely to contribute to the decrease of precipitation during the New Year's holiday.

### 3.5 Water vapor budget and atmospheric circulation

Our analysis demonstrates that in association with the significant precipitation reduction over southern China, anomalous

negative departure of the SH occurs. One further question is that what causes such water vapor deficits during the New Year's holiday. Here, we discuss the possible factors relevant to the atmospheric column water vapor balance as well as their individual contributions. Regional changes of the column water vapor are essentially related to two components. One is the budget between evaporation and precipitation, the other is the budget of moisture inflow and outflow from the horizontal air motion crossing the four lateral boundaries. We investigate the corresponding anomalies separately during the holidays.

First, we computed the differences between the evaporation and the total precipitation (evaporation minus precipitation, i.e., E-P) using the ERA-Interim data from 00:00 and 12:00 UTC. Note that both evaporation and precipitation are forecasted, and the accumulated variables are collected at step 12. We analyzed the E-P anomaly (mm, equivalent to kg m$^{-2}$) during the holidays (figure not shown). From days +1 to +5, the E-P anomalies were all positive values with a mean of 0.85 mm. In addition, the anomalies on days +3, +4 and +5 were all significant at the 0.05 level. The maximum of the E-P anomaly

(+1.25 mm) appeared on day +4. We also investigated the evaporation independently and found that the evaporation is enhanced during holidays, with the mean of the evaporation anomalies reaching +0.39 mm (figure not shown). We should note that the positive E-P anomalies occur concurrently with precipitation reductions. The E-P anomalies are likely dominated by precipitation changes. Although the positive E-P anomaly implies a net gain, this effect cannot account for the total net loss of the column water vapor.

Second, we investigated the column water vapor budget contributed by horizontal transport. Here, we simply estimated the moisture flux across the borders (i.e., the box in Figure 1). The vertically integrated moisture transport ($Q$) crossing the border is defined as follows:





$$Q = \int_L \left[ \frac{1}{g} \int_{500hPa}^{1000hPa} q \cdot V dp \right] dl , \qquad (2)$$

where $V$ is the horizontal wind vector, g is the gravitational acceleration, $q$ is the SH and $L$ is the length of the border. Moisture transport above 500 hPa is ignored as its contribution to the total column moisture transport is relatively small, and large winds tend to cause errors in the upper troposphere where the SH is quite small (Qian et al., 2009). Along the northern

and southern borders, we computed the meridional transport, while along the eastern and western borders, we computed only the zonal transport. The summation of the meridional and zonal transports yields the net budget. The result shows the net water budgets are negative from day +1 to day +5 (figure not shown), with a mean of $-2.9 \times 10^7$ kg s$^{-1}$. Note that the anomalies of days +1, +3 and +4 are all significant at the 0.05 level. The lowest water vapor budget is $-4.4 \times 10^7$ kg s$^{-1}$ (about -1.96 kg m$^{-2}$ day$^{-1}$), occurring on day +4. This simple calculation suggests that the horizontal moisture transport dominates the

anomalous variations of the total net water vapor budget.

Because the negative anomaly in the water vapor budget may be caused by a reduced inflowing water vapor transport or by an increased outflowing transport, we investigated the transport crossing of each of the boundaries during the holidays. We found that the zonal transports crossing the eastern and western borders are negligible (figure not shown). In contrast, the water vapor transport at the southern boundary has the greatest magnitude (Figure 9). The southern border shows continuous

negative anomalies from day +1 to day +5, and the largest departure, of $-5.5 \times 10^7$ kg s$^{-1}$, occurs on day +4. The anomalies during days +2 to +5 are significant at the 0.05 level and have a mean of $-4.4 \times 10^7$ kg s$^{-1}$. Thus, the negative transport crossing the southern border implies an enhanced outflow, and is largely responsible for the net loss of the column water vapor during holidays.

The water vapor transport over this large area should be closely related to the regional atmospheric circulation. Previous

studies have emphasized the importance of the anomalous atmospheric circulation over the western Pacific in modulating the precipitation over southern China by influencing the moisture transport and convergence (e.g., He et al. 2007; Li et al., 2013; among others). To elaborate on the details of the regional atmospheric circulation around the New Year's holiday, we investigated the tropospheric horizontal winds over eastern Asia. The vertical mean horizontal winds averaged from 1000 hPa to 700 hPa are plotted in Figure 9b. Over southeastern China, there are clear dominant northerly wind anomalies. The

northerly winds flow toward the east over the western Pacific, south of approximate 25°N, and then turn northward. Consistent with this cyclonic circulation anomaly, eastern China and the neighboring western Pacific experience remarkable reductions in the column water vapor. This pattern remains stable when looking at each of the pressure levels in the mid-troposphere. For example, at the 850 hPa level, the anomalous cyclone is more evident and the strong northerly winds over southeastern China are greater than the vertical means (Figure 9c). Interestingly, the precipitation reduction in the southern

China is well captured in the ERA-Interim reanalysis, and such a reduction is physically consistent with the large-scale atmospheric circulation change. We repeated the composite analysis of the atmospheric circulation using the NCEP/NCAR Reanalysis I and NCEP–DOE Reanalysis II datasets and found a similar pattern (figures not shown). Employing the same ERA-Interim data but a shorter data length (2001-2012), Gong et al. (2014) also reported a similarly anomalous circulation



pattern over East Asia around the Chinese New Year. This anomalous cyclone is likely a robust signal of the holiday weather. Generally, this anomalous cyclone plays a dominant role in bringing stronger northerly wind, causing drier humidities and lower water vapor values, finally resulting in less precipitation over southeastern China.

**4 Discussion of possible aerosol effects**

**4.1 Air pollution reduction over eastern China**

The CSF is a cultural tradition that is directly related to the daily activity of human beings, particularly economic activities. The concurrent phenomena observed in atmospheric physics and weather likely imply links between these systems. We suspect the holiday anthropogenic aerosols act as an intermediate connection. Aerosol concentrations were widely reported to considerably decrease during this holiday. For example, Tan et al. (2009) investigated the 1994-2006 observations from 13 monitoring sites around the Taipei metropolitan area and found that the concentrations of $NO_x$, CO, nonmethane hydrocarbon, $SO_2$ and $PM_{10}$ were lower during the New Year period than during the non-New Year periods. Using three different approaches considering the thermal power generation, satellite retrieval products and statistical and modeling attributing, Lin and McElroy (2011) estimated that the economic slowdown during the celebrations of CSF was responsible for a notable reduction in anthropogenic emission. According to their estimations for 2005, 2007 2008 and 2010, the CSF contributes a $NO_x$ emission reduction of 12%. In addition, the 2009 CSF contributed a reduction of 10%. These estimations are comparable to those given in the work of Gong et al., (2014), in which they reported a $PM_{10}$ reduction derived from 323 surface station measurements over eastern China and estimated the magnitude of the $PM_{10}$ concentration decrease to be approximately -9.24% for the days -4 to +5 (excluding day 0 to rule out the New Year's Eve firework emissions).

Note that the boreal winter is the season when the air pollution concentration reaches its peak over China. Thus, we investigated the climatic mean and holiday changes of the aerosol optical depth (AOD). Here, the AOD data were obtained from the Moderate Resolution Imaging Spectroradiometer (MODIS) data sets produced by using the AOD 550 Dark Target Deep Blue algorithm and combined by the MOD08_D3 and MYD08_D3 product (Platnick et al., 2015a; Platnick et al., 2015b). These daily level-3 gridded atmospheric data are available from 2002 to 2012 at a 1-degree spatial resolution. As shown in Figure 1, the climatic mean AOD during January and February for eastern China is remarkably greater than those of the surrounding areas and has two maximum centers, one appearing to the north (i.e., the northern plains) and the other appearing to the west (namely, over Sichuan Basin). Concurrently, a considerably high AOD appears over southern China, ranging from 0.4 to 0.8. For comparison, the spatial distribution of $PM_{10}$ concentrations in January-February is also provided. Generally, the $PM_{10}$ concentration in the north is higher than that in the south. However, the pollution in southern China is also severe in the winter.

To better understand the aerosol changes during the holidays, we simply investigated the frequency distribution of the AOD for days [+1, +5]. During this period, a significant net water vapor reduction is observed. Because frequent cloudy weather causes a smaller clear-day AOD sample, it is difficult to calculate a meaningful daily AOD anomaly. Alternatively,



we selected days [-15, -11] to compute the preholiday period AOD frequency distribution for comparison. The frequencies for the holiday and preholiday periods were both calculated based on the original AOD data using a bin width of 0.2 (Figure 10). The ambient atmosphere during the holidays is much cleaner than that of the preholiday period. On average, the AOD means for days [-15, -11] and days [+1, +5] are 0.67 and 0.58 for southern China, respectively. During the holidays, the

magnitude of AOD reduction is shown to be approximately -13.4%. The improvement of air quality is well demonstrated by the frequency changes shown in Figure 10. The frequencies of the cases in which AOD > 0.6 are 50.4% and 36.6% for days [-15, -11] and days [+1, +5], respectively. This result shows a decrease of 13.8%. Note that the largest frequency reduction (approximately -6.7%) occurs in the bin of 0.6-0.8. Additionally, the daily $PM_{10}$ data were also analyzed for comparison and were calculated from the daily air pollution index (API) (Gong et al., 2007). There are 113 stations in our study area. Similar

to the AOD, the $PM_{10}$ frequency distribution was calculated with a bin width of 20 µg m$^{-3}$ for both the holidays and preholidays. The frequency of $PM_{10}$ concentrations greater 100 µg m$^{-3}$ during the holiday period is 19.8%, while the frequency for the same bin during the preholiday period is 41.7%. There is a 21.9% reduction in the high-$PM_{10}$ concentration cases during the CSF holidays. Not only do the frequencies change but the mean $PM_{10}$ concentrations also evidently decrease. The mean $PM_{10}$ concentrations for days [-15, -11] and days [+1, +5] are 98.9 µg m$^{-3}$ and 71.5 µg m$^{-3}$, respectively.

In addition to southern China, eastern Asia shows a marked drop in air pollution concentrations (figure not shown). We investigated the mean AOD and $PM_{10}$ values of eastern China (108°E-123°E and 21°N -42°N). The AOD values decrease from 0.60 during days [-15, -11] to 0.58 during days [+1, +5]. The frequency of cases with AOD > 0.6 is 33.3% during the holiday period, while the frequency for the preholiday period is 38.1%. Additionally, the mean $PM_{10}$ concentrations for days [-15, -11] and days [+1, +5] are 112.7 µg m$^{-3}$ and 85.2 µg m$^{-3}$, respectively. The reduction in high-pollution cases is notable.

The proportion of $PM_{10}$ >100 µg m$^{-3}$ records declined from 47.9% to 30.1%. The simultaneous and anomalous variations of the pollution concentrations of $PM_{10}$ in southern and eastern China can be clearly seen in their temporal composites, as shown in Figure 10c. The significant negative anomalies around the LNYD clearly appear in both eastern and southern China, and their lowest departures both occur a couple of days before New Year's Day. For eastern China, the mean value for days [-4, -1] is -12.26 µg m$^{-3}$, which is approximately 14.71% smaller than the climatic background concentration (83.35

µg m$^{-3}$). In southern China, the mean of the anomalies during days [-4, -1] is -9.07 µg m$^{-3}$, being 12.63% lower than that of the background (71.84 µg m$^{-3}$). These robust AOD and $PM_{10}$ signals clearly demonstrate that around the LNYD, eastern China experiences a relatively cleaner ambient atmospheric environment although the climatic background air pollution level in January and February is the highest of all seasons.

## 4.2 Time-lag correlation between $PM_{10}$ concentration and the anomalous cyclone

Generally, aerosols might influence regional precipitation in two aspects. The first is via the indirect effects. Secondly, aerosols may also influence regional precipitation by altering the thermal features and further modulating the regional atmospheric circulations. The involved aerosol-cloud-precipitation and aerosol-circulation-precipitation interactions/feedbacks make it hard to clearly identify aerosol impacts on precipitation. There are increasing numbers of



works addressing this issue. For example, Zhao et al. (2006) analyzed 40-years' observations of precipitation, satellite aerosol data and meteorological sounding data over eastern China and suggested a positive feedback mechanism (i.e., more aerosols cause less precipitation, which in turn leads to more aerosols) that will induce an acceleration of the reduction of precipitation in eastern China. Particularly, the observed changes of light rains are likely related to aerosols. Qian et al.

(2009) employed long-term observational data from 1956-2005 and found that both the frequency and amount of light rain have decreased over eastern China. Their numerical simulations suggest that the light precipitation tends to be suppressed due to a higher cloud droplet number condensation and smaller droplet sizes under polluted conditions. Some studies suggested that aerosols induce surface cooling and weaken land-sea thermal contrast, thereby weakening monsoon circulations and the resultant precipitation changes (e.g., Wang et al., 2013; Ye et al., 2013; Guo et al., 2013; among others).

The above studies and mechanisms put an emphasis on climatic scale processes. We note a few studies addressing the synoptic scale circulation anomalies caused by aerosol radiative effects. For example, Jones et al. (2004) suggested the dust aerosol heating in northern Africa plays an important role in changing the amplitudes of the downstream easterly waves over a couple of days. Based on observational data, Ding et al. (2013) reported that the pollution caused by agricultural burning and fossil fuel combustion in some provinces over East China results in significant modification of regional daily

temperature and precipitation through pollution-boundary layer dynamics and aerosol-radiation-cloud feedbacks. In addition to the direct and semi-direct radiation impacts on meteorology, numerical experiments with cloud microphysical processes show that Asian pollution invigorates weather activities along the North Pacific storm tracks (Zhang et al., 2007; Wang et al., 2014).

Is there physical linkage between the holiday aerosol reduction and anomalous precipitation (as well as the

precipitation-relevant variables, such as the humidity, cloud, and temperature)? With respect to the indirect effects, the increases of anthropogenic aerosol concentration may enhance cloud lifetime and delay precipitation (e.g., Albrecht, 1989). Meanwhile, the cloud droplet size might be smaller, which tends to lead to ineffective coagulation precipitation (e.g., Rosenfeld et al., 2008). During the CSF holidays, however, the observed aerosol reduction accompanies a cloud cover reduction, a lower precipitation frequency and smaller precipitation amount. Identifying each role played by these cloud

microphysical processes remains difficult. More importantly, note that in the observations, the precipitation reduction is strongly related to the significant dry of the SH, which is caused by anomalous northerly winds. In fact, an anomalous cyclone dominates East Asia and the western Pacific region during the CSF holidays (Figure 9). The anomalous northerly winds over southern China are just located in the rear side of the anomalous cyclonic circulation. We speculate that the anomalous cyclone is likely related to the aerosol reduction around the holiday period.

Note that there is a time lag between the $PM_{10}$ reduction and the anomalous cyclone. Gong et al. (2014) reported that during days [-4, 0], there is no anomalous cyclone over East Asia, but at the same time, the temperature cools significantly. These phenomena can be observed in both the long-term (1979-2012) and short-term (2001-2012) data periods. The anomalous cyclone appears after New Year's Day in the troposphere, moves eastward, and disappears after approximately 12 days. This result is highly consistent with our analysis, as demonstrated in Figure 9. The largest aerosol reduction occurs



on days [-3, -1]. At the same time, the greatest temperature cooling in eastern China was observed (c.f., Gong et al., 2014, figures 2 and 4). The anomalous cyclone is likely triggered/enhanced by the cooling of the low-middle troposphere, which favors constructive baroclinic interaction between the upper and lower troposphere and enhances the development of the midlatitude cyclone system.

To address this issue, we further analyzed the time-lag correlation between the holiday cyclone and the preceding $PM_{10}$ concentrations. Because the anomalous cyclone is most robust during days [+4, +6], we selected its corresponding center (140°E-145°E and 25°N-35°N) to calculate a regional mean for the 500 hPa height. Then, we computed its correlation with the daily mean $PM_{10}$ anomalies over eastern China for each of day around the LNYD. The daily $PM_{10}$ data were taken from Gong et al. (2014). Note that the $PM_{10}$ observations are highly variable. To reduce the possible uncertainty of the correlation,

we employed only those $PM_{10}$ stations with >70% of the daily data available during the period from 2001-2012. To suppress the noise and identify a stable signal, we further used a 3-day window when computing the means, i.e., the data on a specific day as well as those of the day before and after are averaged. On New Year's Eve, there is a sharp $PM_{10}$ peak due to intense firework burning (Figure 10c). Therefore, the data on New Years' Day are intentionally excluded. The results are plotted in Figure 11; note that a time-lag of 0 refers to day +5, a time-lag of -2 refers to day +3, etc. The simultaneous correlation

between the regional mean $PM_{10}$ and the cyclonic pressure is quite weak (only -0.2). Interestingly, their correlations increase substantially as the $PM_{10}$ leading time grows, until approximate 10 days. The strongest correlations occur when the $PM_{10}$ leads by -9 to -6 days, with correlation coefficients varying from +0.52 to +0.57. The maximum appears for a time-lag of -9. All these correlations are statistically significant at the 0.1 level. As indicated previously, around the holidays, the $PM_{10}$ experiences similar changes over eastern China. Such a time-lag relationship should be found for the subregional $PM_{10}$ data.

We repeated the cross-correlation analysis using only the mean $PM_{10}$ values over southern China. The largest correlations also occur when the $PM_{10}$ values lead by 6 to 9 days, as expected. Note that 9-day and a 6-day leading times correspond to days -4 and -1, respectively. During these days, there are remarkable $PM_{10}$ reductions (Figure 10c). Their significant positive correlations suggest that when the $PM_{10}$ concentration is lower during days [-4, -1], the intensity of the anomalous cyclone on days [+4, +6] over the western Pacific is likely to be stronger. Thus, the anomalous cyclone is significantly related to the

continental $PM_{10}$ concentration that occurred approximate one week previously.

     The aerosols are likely to cause the circulation anomalies by affecting the atmospheric thermodynamics. We computed the cross-correlations of the regional mean temperatures with the anomalous cyclone on day +5. In eastern China, there are 394 available meteorological stations. We averaged these stations' daily temperature anomalies to derive a regional mean time series and then computed its correlation with that of the 500 hPa height on day +5. The results show that the cyclone is

closely related to the temperature anomalies that occurred approximate 5 days previously. A maximum correlation of 0.52 was observed when temperature leads the pressure by 4 to 5 days. We also computed the cross-correlations using the 31 $PM_{10}$ collocate stations. The features are almost identical (Figure 11). Similar to the $PM_{10}$ reduction, the holiday temperatures cool over eastern China. Therefore, the regional temperature over southern China experiences a similar cross-correlation with the following cyclone (figure not shown). These results demonstrate that a stronger cyclone over the western



Pacific on days [+4, +6] is often accompanied by a lower preceding $PM_{10}$ concentration and cooler temperatures over eastern China that occurred approximate one week previously.

Cyclones in the middle latitudes generally emerge and develop over the course of about one week, moving eastward. Meanwhile, the cold advection behind the pressure trough plays an essential role during the cyclones' developing stage. 5 Such advection might be contributed to by anomalous horizontal winds or/and by temperature gradients. As shown in Figure 12, the lower-middle troposphere shows no evident northerly wind anomaly over East Asia and the western Pacific before New Year's Day. However, there is a significant cooling in the lower-middle troposphere, as indicated by the negative 1000-500 hPa thickness anomalies during days [-4, 0]. The cooling center is located between 100 E-130 E and 30 N-40 N. In this case, the negative temperature anomalies can result in anomalous cold advection due to the climatic northerly winds. Thus, 10 the enhanced cold advection could reinforce the baroclinic system. Under such circumstances, an anomalous cyclone will be enhanced or triggered (Hakim et al., 2003). The cyclone brings anomalous northerly winds to eastern Asia, reducing the atmospheric SH and, consequently, resulting in less precipitation over southern China. In addition, the northerly winds would help reduce the $PM_{10}$ concentration and can cause a colder temperature. We think that when considering the simultaneous temperature changes in association with short period aerosol anomaly, the atmospheric feedback would be 15 ignorable. However, during a moderate period (such as >3 days to one week) the atmospheric feedback is likely discernible. In Figure 11, we show the correlation peak of the temperature lagging the $PM_{10}$ by a couple of days, which is evidence of such a feedback.

## 5 Summary

Briefly, the major findings of our analysis are summarized as follows:

20 The long-term station precipitation data from 1979 to 2012 were analyzed with a focus on the possible changes during the CSF holidays. We found that the precipitation frequency over southern China experiences a significant holiday reduction. The largest reduction occurs on days [+4, +6], with a mean anomaly of -7.4% and a minimum anomaly of -9.4% on day +5. At the same time, the daily precipitation amounts from day +2 to day +5 show continuous negative anomalies, and their mean is -0.62 mm $d^{-1}$. The Monte-Carlo test implies that the holiday-related frequency and amount anomalies are 25 significantly different from random occurrences. The spatial distributions of the mean frequency anomalies for days [+4, +6] clearly show that a reduction appears across the whole southern China. Among the 155 stations, negative anomalies were observed at 148 stations.

The holiday precipitation anomalies are strongly linked to the relative humidity and cloud cover. The station ΔRH shows an evident decrease from day +2 to +7, and the lowest anomalies appear on days [+4, +6], with a mean of -3.9%. 30 When all precipitation days excluded the ΔRH shows similar decreases, the lowest values also occur on days [+4, +6], with anomalies of -0.97%, -0.69% and -0.38%, respectively. The ΔRH vertical profile demonstrates a significantly drying under approximately 800 hPa. The ERA-Interim reanalysis data revealed that the negative anomaly at 850 hPa is -3.9% and that at



1000 hPa is -4.2%. The mean ΔRH of layers between 850 hPa and 1000 hPa is -3.9%. The negative anomalies in the lower troposphere are consistent with the significant decreases in the LCC. The daytime station LCC shows an evident reduction during days [+4, +6], with a mean of -6.1%. Meanwhile, the daily ERA-Interim LCC also displays a notable reduction during days [+4, +6]. The corresponding mean is -5.0%, and a minimum of -5.9% appears on day +5. This magnitude as well

as its spatial distribution are comparable to that of the station-based daytime LCC anomaly.

The anomalous relative humidity is found to be mainly caused by a drying of the water vapor in the lower-mid troposphere over southern China during the holidays. Evident negative SH anomalies persist from days -3 to +7 in the station observations. The lowest values appear on approximately days [+4, +5]. The average ΔSH for days [+4, +6] is -0.73 g kg$^{-1}$. When the precipitation days are excluded, the ΔSH shows a similar continuous reduction after New Year's Day. A minimum

of -0.54 g kg$^{-1}$ appears on day +4, while the mean for days [+4, +6] is -0.46 g kg$^{-1}$. Significant drying of the water vapor is observed for the whole low troposphere below 700 hPa. Below 800 hPa, the ΔSH values are all below -0.50 g kg$^{-1}$. The mean value between 800 hPa and 1000 hPa is -0.70 g kg$^{-1}$. The reduction of the water vapor in the lower-middle troposphere likely plays the dominant role in causing anomalous relative humidities, low-level clouds and precipitation. This water vapor deficit results from the anomalous meridional horizontal moisture transport. During the holidays, a large-scale cyclonic

circulation appears over the western Pacific, which brings anomalous northerly winds to East Asia and leads to negative water vapor flux in the troposphere.

We suppose that holiday aerosol is likely to play a role in generating anomalous cyclones. As the observation $PM_{10}$ and satellite AOD demonstrated, the aerosol concentrations around the CSF decrease evidently. In East Asia, the largest reduction of $PM_{10}$ occurs on days [-3, -1]. A simultaneous cooling was observed (e.g., Gong et al., 2014). Cross-correlation

demonstrates that approximately one week after a lower $PM_{10}$ concentration and a cooler temperature over eastern China, a stronger cyclone is observed over the western Pacific. The cooling is likely to lead to anomalous cold advection, which enforces the constructive baroclinic interactions between the upper and lower troposphere, thus triggering or enhancing the development of a midlatitude cyclone system. The cyclone brings anomalous northerly wind to East Asia, reduces the atmospheric SH, and consequently results in less precipitation over southern China. To what extent the cooling is contributed

by the direct/indirect effects of aerosol and the role played by the atmospheric feedback are still unclear. Whether these physical/chemistry processes depend on the aerosol emission region and the aerosol types, needs further clarification by elaborate observation analysis and modeling.

**Acknowledgements.** This research was supported by projects NSFC-41621061, and NSFC-40975043. The meteorological data were obtained from the China Meteorological Administration. ERA-Interim reanalysis data used in this study were obtained from ECMWF at http://www.ecmwf.int/. The API data were obtained from the Chinese Ministry of Environmental Protection. Satellite data used in this study were obtained from MODIS Web of NASA at https://modis.gsfc.nasa.gov/.





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



**Table 1:** The Lunar New Year's Days from 1979 to 2012

| Year | Date | Year | Date |
|------|------|------|------|
| 1979 | 28 January | 1996 | 19 February |
| 1980 | 16 February | 1997 | 7 February |
| 1981 | 5 February | 1998 | 28 January |
| 1982 | 25 January | 1999 | 16 February |
| 1983 | 13 February | 2000 | 5 February |
| 1984 | 2 February | 2001 | 24 January |
| 1985 | 20 February | 2002 | 12 February |
| 1986 | 9 February | 2003 | 1 February |
| 1987 | 29 January | 2004 | 22 January |
| 1988 | 17 February | 2005 | 9 February |
| 1989 | 6 February | 2006 | 29 January |
| 1990 | 27 January | 2007 | 18 February |
| 1991 | 15 February | 2008 | 7 February |
| 1992 | 4 February | 2009 | 26 January |
| 1993 | 23 January | 2010 | 14 February |
| 1994 | 10 February | 2011 | 3 February |
| 1995 | 31 January | 2012 | 23 January |




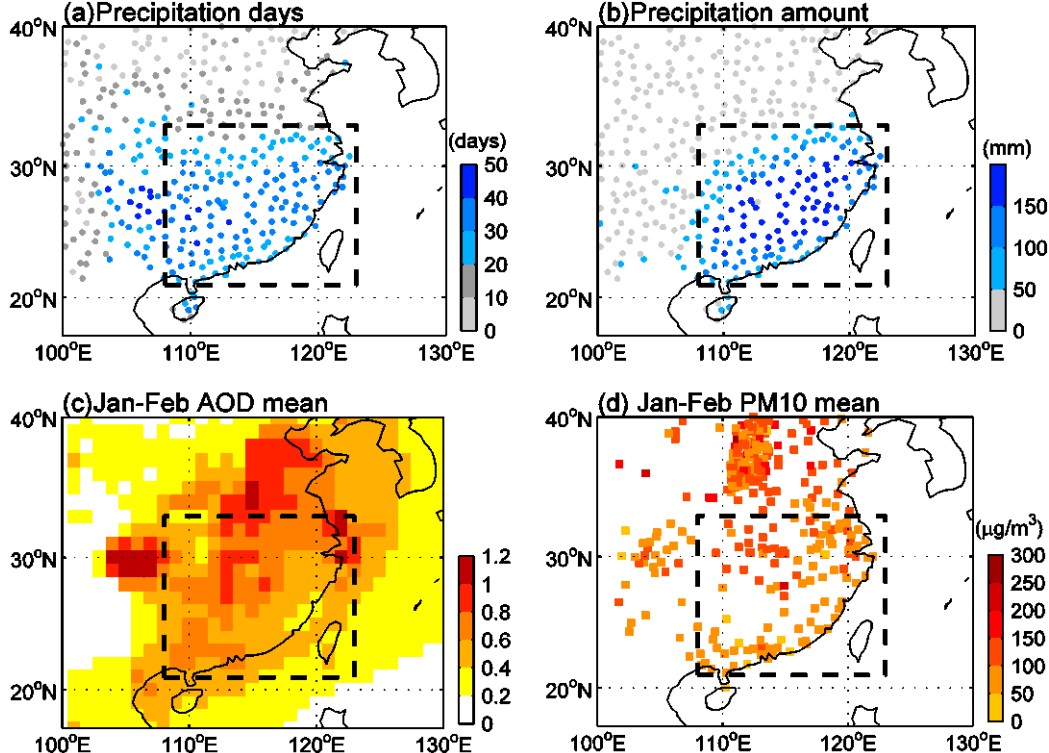

**Figure 1:** Spatial distribution of January- February precipitation days (a) and precipitation amount (b) for the period of 1979-2012. Spatial distribution of the January- February averaged Aerosol Optical Depth (AOD) for the period of 2002-2012 (c) and PM$_{10}$ concentration for the period of 2001-2012 (d) over eastern Asia. The study area is shown in a black dashed rectangle.





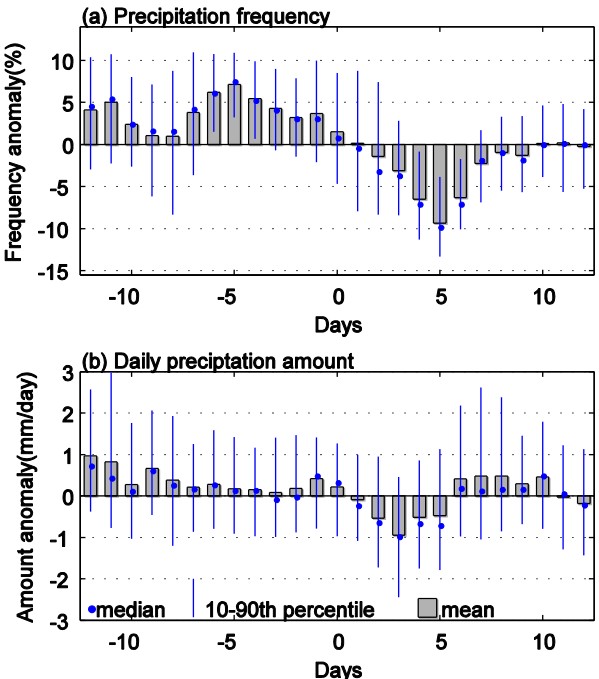

**Figure 2:** The anomalies of precipitation frequency (a) and amount (b) computed based on 155 stations. The means are shown as gray bars, and the median; and the range of the 10-90th percentiles are also plotted for comparison.





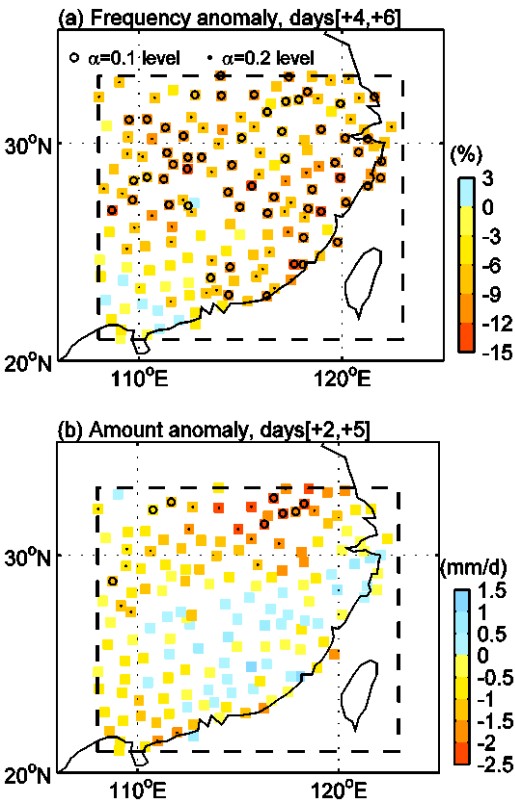

**Figure 3:** Anomalies of the precipitation frequency (a) and amount (b). The significances are estimated using a Monte-Carlo approach; stations with circles and dots denote that all days have values significant at the 0.1 and 0.2 levels, respectively.



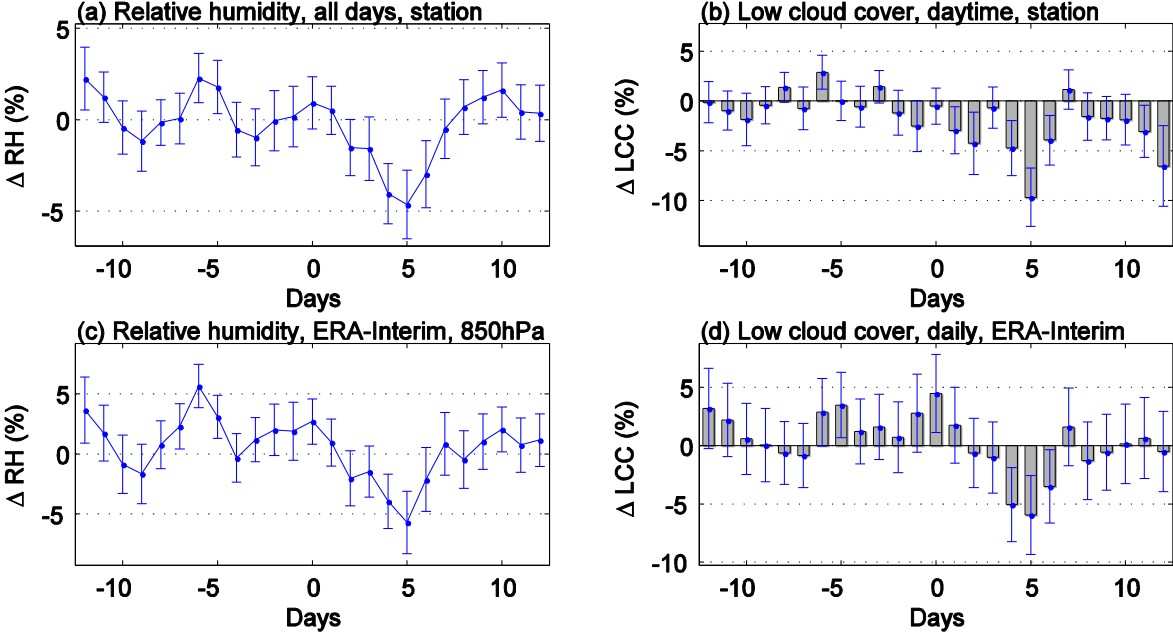

**Figure 4:** Station observational relative humidity anomalies (ΔRH) estimated from all available days (a) and the daytime low cloud cover (LCC) anomalies (b) during the holidays from 1979-2012. ERA-Interim ΔRH at 850 hPa (c) and the daily LCC anomalies during the holidays. Standard error is shown as error bars.




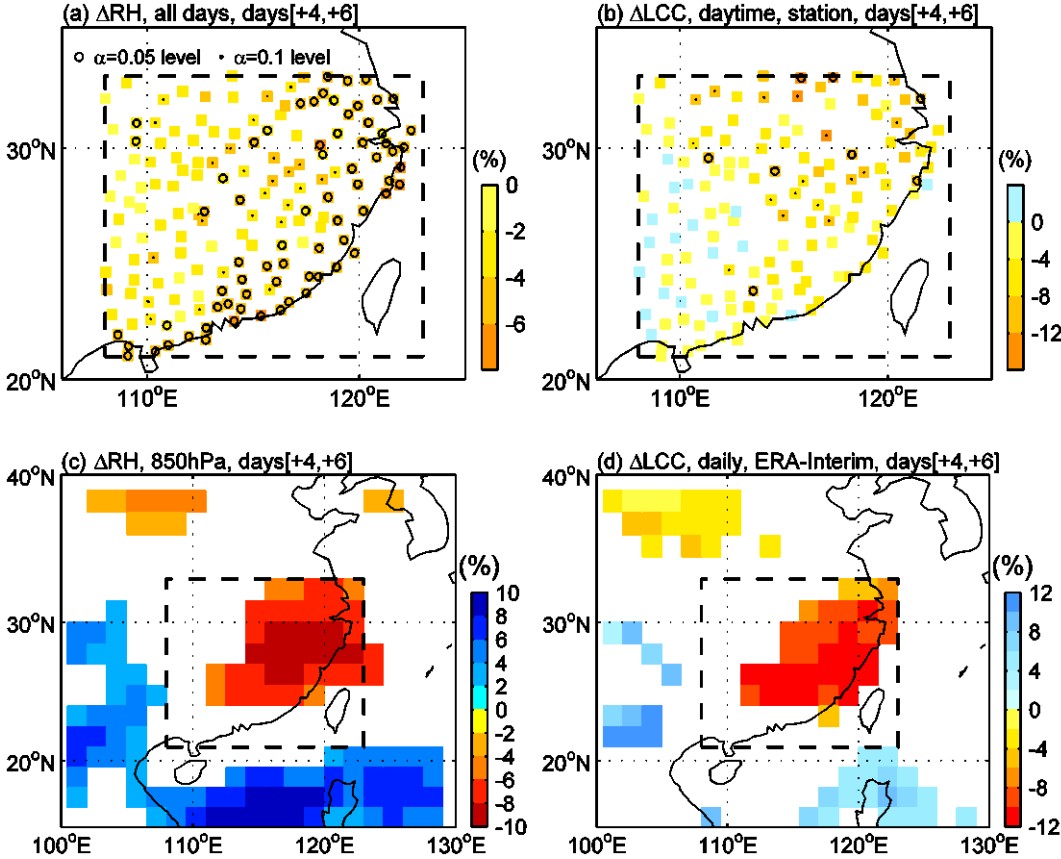

**Figure 5:** Station anomalies of relative humidity estimated from all available days (a) and daytime low cloud cover (b) during days +4 to +6. The significance is estimated using a *t*-test; stations with circles and dots denote values are significant at the 0.05 and 0.1 level, respectively. Spatial distribution of ERA-Interim ΔRH at 850 hPa (c) and the daily LCC anomalies (d) during days +4 to +6. Only the significant values (at the 0.1 level) are plotted.



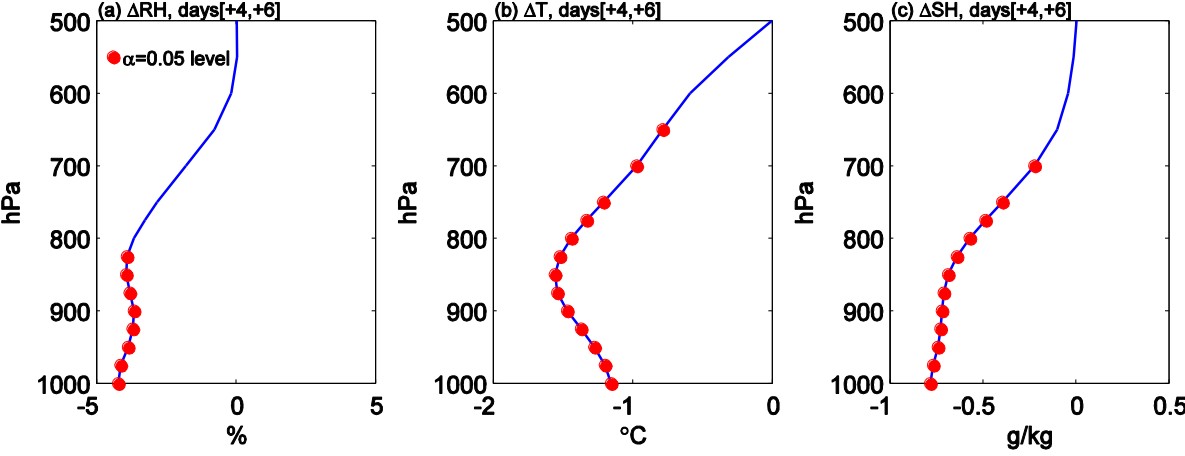

**Figure 6:** The vertical profiles of the relative humidity (a), temperature (b) and specific humidity (c) anomalies below 500 hPa. The significance level is estimated using a left-tail *t*-test, and the values significant at the 0.05 level are indicated with red dots.



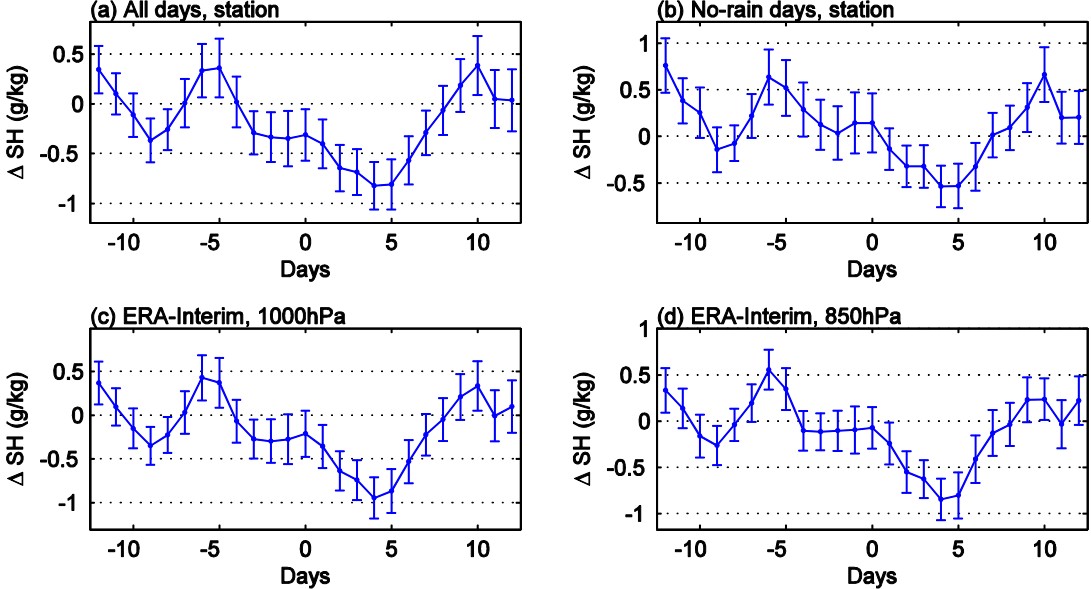

**Figure 7:** Means of station-specific humidity anomalies (ΔSH) during the holidays from 1979-2012 as estimated from all available days (a) and no-rain days (b). The corresponding ERA-Interim ΔSH at 1000 hPa (c) and 850 hPa (d) are also plotted.





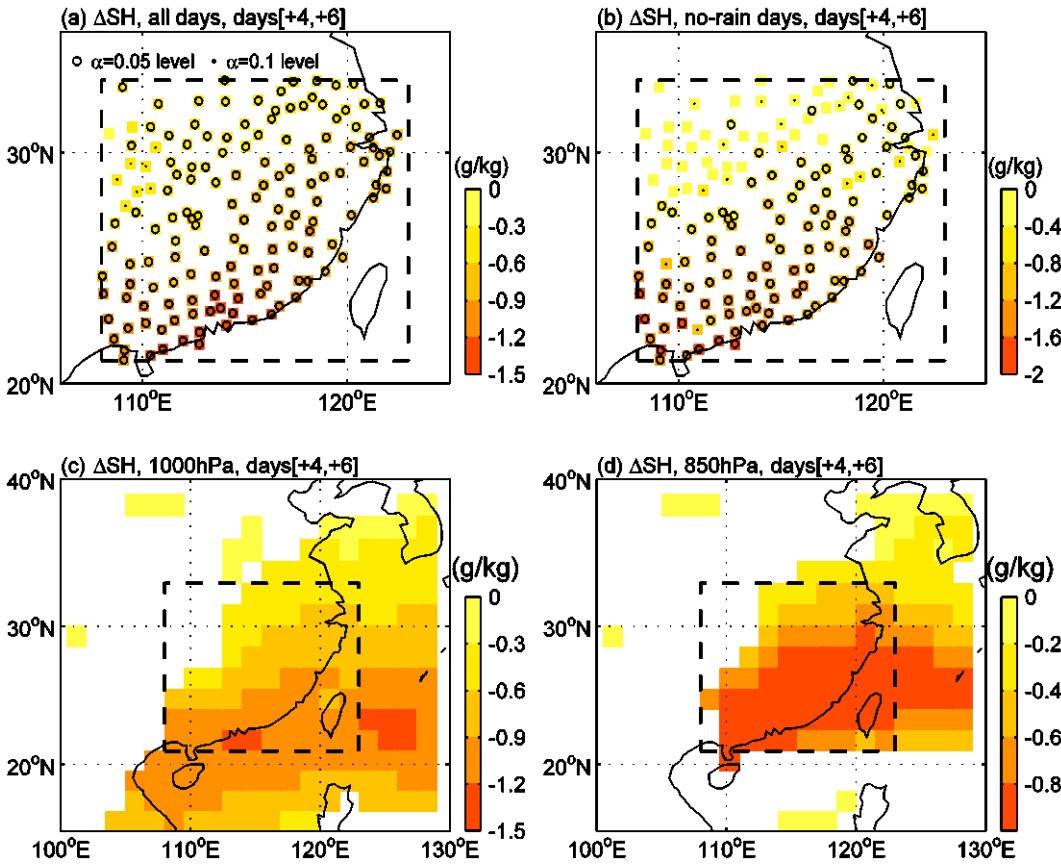

**Figure 8:** Spatial distribution of the observational specific humidity anomalies (ΔSH) during days [+4, +6] as estimated from all available days (a) and no-rain days (b). The spatial distributions of the ERA-Interim ΔSH at 1000 hPa (c) and 850 hPa (d) during days [+4, +6]. Only the significant values (above the 0.05 level) are plotted.





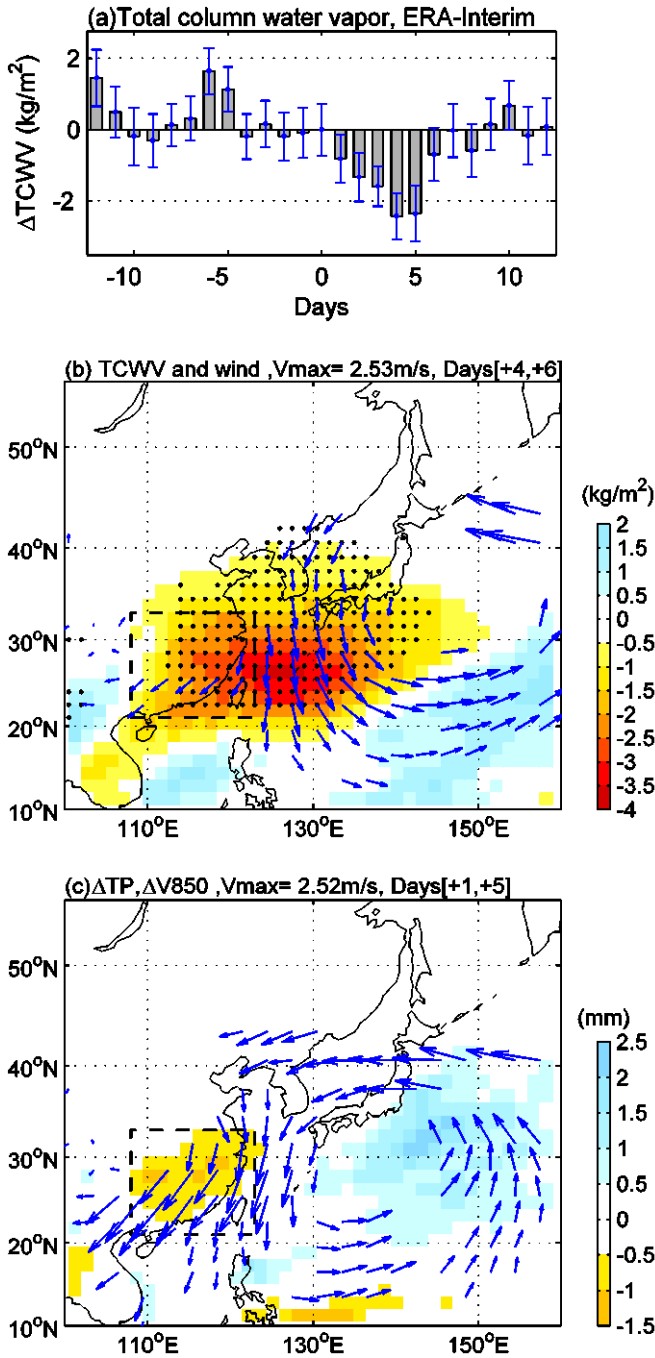

**Figure 9:** (a) Temporal anomalies of the total column water vapor (ΔTCWV) during the holidays. (b) Spatial distribution of the ΔTCWV (in color shading with unit of kg m$^{-2}$) and the 700-1000 hPa mean horizontal winds (in vectors with unit of m s$^{-1}$) during days [+4, +6]. Stipples denote significant ΔTCWV (at the 0.05 level). Only the anomalous wind vectors significant at the 0.05 level are plotted. (c) Spatial distribution of the mean anomalies of the total precipitation(ΔTP) (in color shading with unit of mm) and the significant (at the 0.05 level) 850 hPa mean horizontal winds (in vectors with unit of m s$^{-1}$) during days [+1, +5].



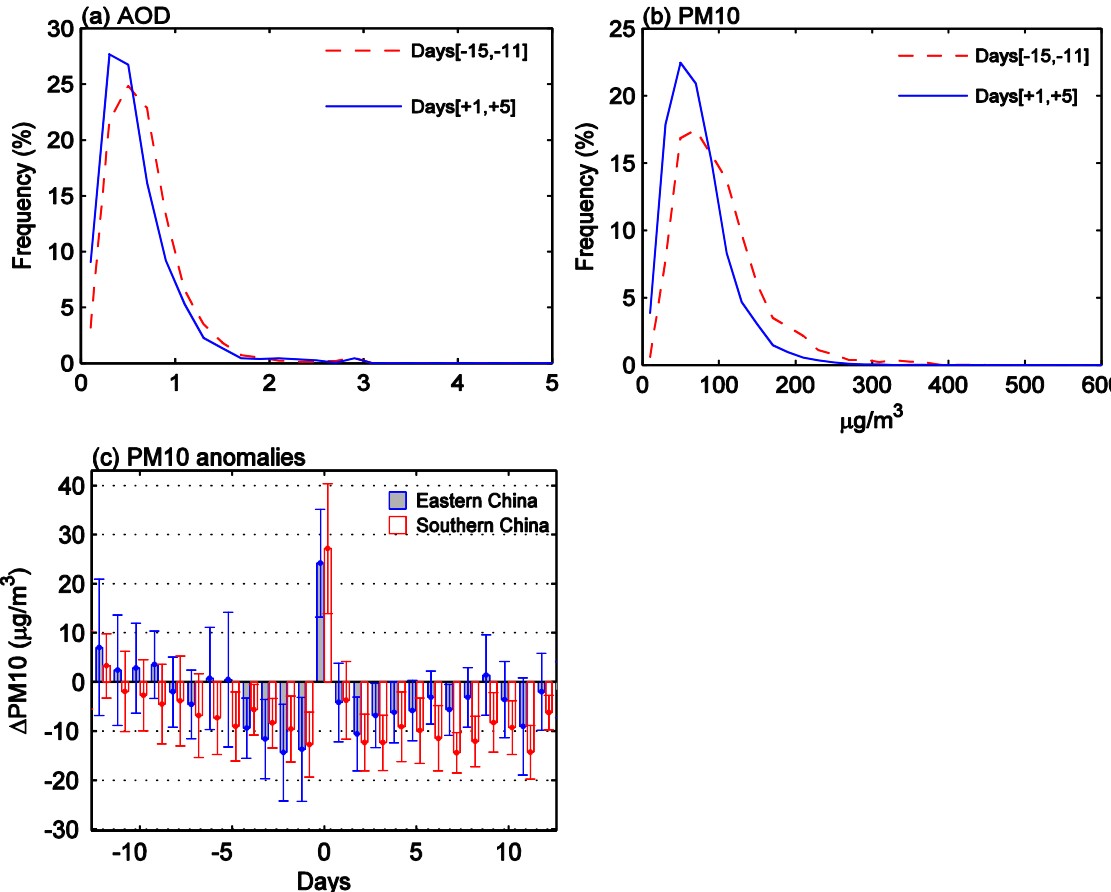

**Figure 10:** Frequency distributions for the AOD (a) and PM$_{10}$ concentration (b) during days [-15, -11] (red dashed line) and days [+1, +5] (blue solid line) over southern China. (c) The temporal anomalies of the PM$_{10}$ concentration in eastern and southern China. Only years with more than 50% PM$_{10}$ station data available are employed for anomalous composites.





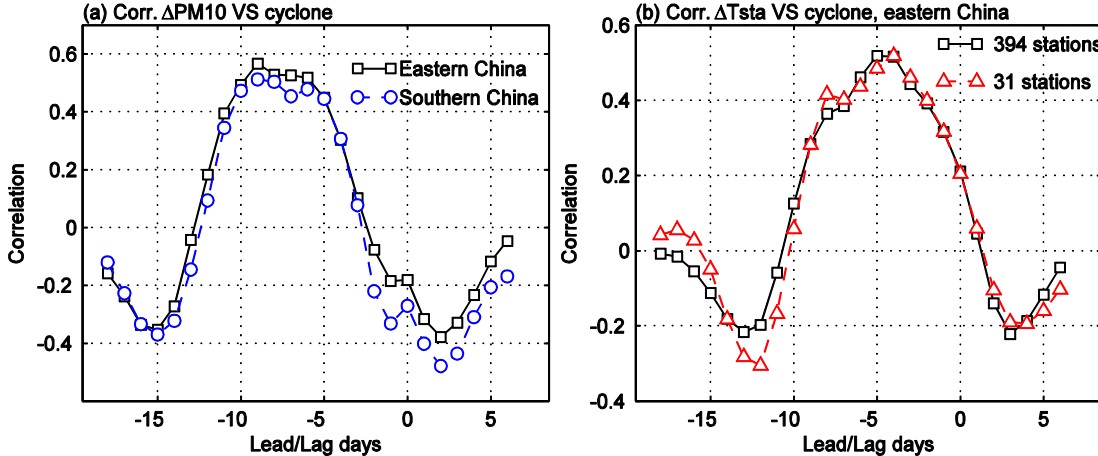

**Figure 11:** (a) Correlations of the 500 hPa geopotential heights over the anomalous cyclone center (140 °E-145 °E and 25 °N-35 °N) on day+5 with the $PM_{10}$ concentration in varying lead/lag days. $PM_{10}$ for eastern and southern China are computed separately. (b) Same as in (a) but for the regional mean temperature over eastern China. Temperatures from all available 394 stations and that from 31 $PM_{10}$ collocated stations are plotted together for comparison.





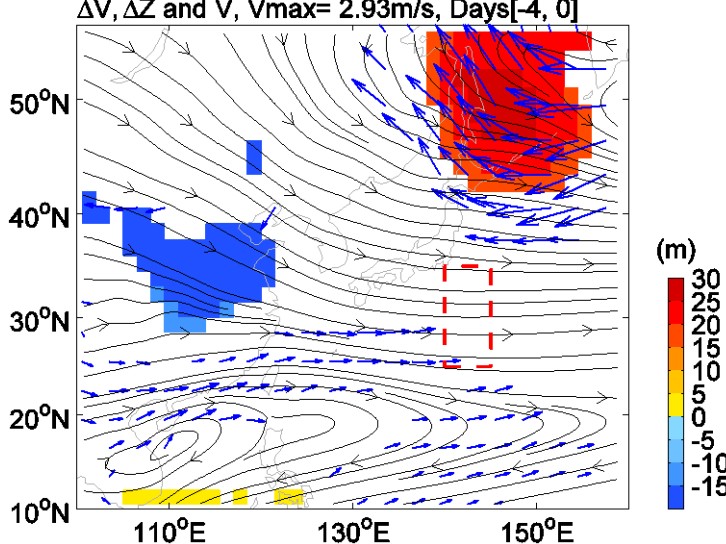

**Figure 12:** The mean horizontal wind anomalies at 1000-500 hPa (in vectors with unit of m s⁻¹) and the 1000-500 hPa thickness anomalies (in color shading) during days [-4, 0]. Only significant (at the 0.1 level) winds and thickness are plotted. The long-term mean horizontal winds between 1000-500 hPa during days [-4, 0] are shown together as the streamlines for comparison. A red dashed rectangle (140 E-145 E and 25 N-35 N) indicates the central location of the anomalous cyclone which is manifest in day+5.