# Peer review of "Anomalous holiday precipitation over southern China"

_Atmospheric Chemistry and Physics, 2018_

## Referee Comment (RC1) · Anonymous Referee #1 · 23 May 2018

This manuscript examined anomalous precipitation changes around the Chinese Spring Festival (CSF) and associated temperature, humidity and circulation changes using extensive station data and the ERA-Interim reanalysis. The results showed that the precipitation tends to decrease during the CSF holiday, and pointed out that the change is mainly caused by the humidity decease associated with an anomalous cyclone circulation. The results are very interesting, especially given that the ERA-Interim data present similar changes in the surface precipitation. However, the authors tried to attribute all these changes to the aerosol decrease due to the economic slowdown without giving persuasive proofs. I cannot agree to this part. The cause-effect relationship between aerosol and precipitation changes cannot be concluded from current results. I suggest a major revision to Introduction and Section 4. Currently they could

be misleading for readers.

1. Page 1, Line 19, 'lower water vapor' → decreased water vapor; Line 21, 'When the precipitation days exclude the mean . . .' this sentense is confusing.

2. The authors emphasized aerosols too much throughout Introduction. Although the authors did not state it clearly, it still feels as if aerosol changes associated with human activities could explain all changes presented in the main text. This is misleading. I suggest the authors emphasize human impacts on weather and climate at diverse spatial/temporal scales rather than aerosols in this section.

3. While this manuscript focused on southern China, are there any changes in precipitation over northern China? Since Gong et al. 2014 showed the cooling during the CSF holiday spreaded over both northern and southern China. It will be better if the authors could give some information on this aspect in the discussion.

4. Page 9, Line 6, what does 'higher' mean here?

5. Page 10, Line 5, and somewhere else in the manuscript, 'medium cloud' → middle cloud

6. Page 11, Line 18, 'plotted Figure 6b'→plotted in Figure 6b

7. Page 16, Line 11, 'The frequency of PM10 concentrations greater' →The frequency of PM10 concentrations greater than

8. As the aerosol loading is greatly increased over East Asia since 1980s, the aerosol loading after 2000 is much larger than that in 1980s. Then, are the aerosol changes shown in Figure 10 dominated by aerosol changes after 2000? Maybe you should normalize the PM10 data for each year before compositing the multi-year mean.

9. The authors examined the time-lag correlation between PM10 concentration and the anomalous cyclone, and found that the correlation is largest if the PM10 leads by -9 to -6 days. Is it possible that this correlation is due to the 1-2-weeks period of

synoptic systems? In other words, the northerlies associated with a synoptic system could decrease the aerosol loading, and it may appears as if the aerosol decrease is correlated with northerlies associated with the next synoptic system that comes in 1-2 weeks. In Figure 11a, the curves rise for positive lead/lag days and may reach a similar height at +10 as that at -9.

---

## Referee Comment (RC2) · Anonymous Referee #2 · 30 May 2018

General Comments:

This manuscript presents the anomalous holiday precipitation over southern China during the Chinese Sprint Festival based on their analysis of the long-term station observations. The associated meteorological parameters are also analyzed to investigate the possible mechanisms of the reduced precipitation. The manuscript is scientifically sound, well organized, written, and concise. I recommend accepting it as minor revision as below.

Specific comments: P3 L24 it is better to use southern China not China since the results are analyzed in southern China in this study. P4 L8-9 What is your criterion to exclude the stations? likely if there is only one missing data do you exclude the site? P4 L29 what is the step 0? P5 L10-18 The statements to calculate the precipitation

frequencies are not clear. Actually how many days do you use, 7 days or 3 days? And it contradicts to the 9 days as found with the aerosol time-lag correlations. P6 L10-15 What do you mean of the specific day here? P6 L30 It looks there are positive departures. P7 L16 section→ subsection P7 L29 could you also shown this sub-region in the Figure? P8 L10 factors-> factor P8 L25 Please give the sample numbers of no rain days P10 L2-3 if total cloud cover shows no evident changes but low cloud covers experiences a significant decrease, does it indicate high cloud covers are increased? Figure 9 is it the horizontal wind or wind anomaly? P16 L1 why do you selected days [-15,-11]? Is it arbitrary? P18 L17 The maximum appears for a time-lag of -9, but aerosol lifetime is generally less than one week? Is there any other mechanism? P20 L19 It is not accurate to use East Asia here.

---

## Author Comment (AC1) · 9 Aug 2018

This manuscript examined anomalous precipitation changes around the Chinese Spring Festival (CSF) and associated temperature, humidity and circulation changes using extensive station data and the ERA-Interim reanalysis. The results showed that the precipitation tends to decrease during the CSF holiday, and pointed out that the change is mainly caused by the humidity decease associated with an anomalous cyclone circulation. The results are very interesting, especially given that the ERA-Interim data present similar changes in the surface precipitation. However, the authors tried to attribute all these changes to the aerosol decrease due to the economic slowdown without giving persuasive proofs. I cannot agree to this part. The cause-effect relationship between aerosol and precipitation changes cannot be concluded from current results. I suggest a major revision to Introduction and Section 4. Currently they could be misleading for readers.

*[Answer] We agree with your opinions. We have modified Introduction and Discussion accordingly. In revision, we generally talked the human activities and the influence on weather and climate, and intentionally to avoid the cause-and-effect relationship between aerosol and weather relation in Introduction. In Discussion, we also only showed PM$_{10}$ anomalies and its time-lag correlation with the atmospheric circulation. Although we have not analysed their cause-and-effect relationship, the diagnostics and correlation analysis should shed lights on the possible mechanism(s) for further studies.*

1. Page 1, Line 19, 'lower water vapor' → decreased water vapor; Line 21, 'When the precipitation days exclude the mean…' this sentence is confusing.

*[Answer] Modified. We changed the sentence 'When the precipitation days exclude the mean…' to 'When the precipitation days are excluded…'.*

2. The authors emphasized aerosols too much throughout Introduction. Although the authors did not state it clearly, it still feels as if aerosol changes associated with human

activities could explain all changes presented in the main text. This is misleading. I suggest the authors emphasize human impacts on weather and climate at diverse spatial/temporal scales rather than aerosols in this section.

*[Answer] Thank you for your suggestion. We modified Introduction, put attention on the diverse human influence on weather and climate. And have avoided to directly mention/attributing aerosols' impact/effect, instead, we simply presented the changes in meteorological variables and air pollution in the context of human activities. When talking the weekly cycles, we mainly introduce what happened in meteorological parameters, and the aerosol changes mentioned occasionally. But when talking the human intensive events, we remain most content of the aerosols/air pollutions, because the relevant studies on the accompanying meteorological changes are very limited.*

3. While this manuscript focused on southern China, are there any changes in precipitation over northern China? Since Gong et al. 2014 showed the cooling during the CSF holiday spreaded over both northern and southern China. It will be better if the authors could give some information on this aspect in the discussion.

*[Answer] As shown in Figure 1a, there are much less precipitation days over northern China than southern China in the winter. The Chinese Spring Festival (CSF) holiday repeatedly occurs every year in January-February across the whole country. During January-February, average precipitation days and amount are 12.8 days and 9.9 mm (means for 319 stations north of 33°N). For the 279 stations south of 33°N, the average precipitation days and amount are 26.5 days and 81.9 mm. Too small precipitation samples in northern China would be difficult to statistically yield a meaningful signal, if any. In Section 2.1 (paragraph 2) of the revision, we have added a sentence to indicate this.*

*In revision, it reads:" In addition, for 319 stations north of 33°N the average precipitation days and amount are 12.8 days and 9.9 mm. Too small precipitation frequency and amount in northern China would be difficult to statistically yield a meaningful signal."*

4. Page 9, Line 6, what does 'higher' mean here?

*[Answer] We have changed the sentence to 'The precipitation reduction could occur with a drier and upper atmosphere'.*

5. Page 10, Line 5, and somewhere else in the manuscript, 'medium cloud' → middle

cloud

*[Answer] All taken.*

6. Page 11, Line 18, 'plotted Figure 6b' →plotted in Figure 6b

*[Answer] Done.*

7. Page 16, Line 11, 'The frequency of PM10 concentrations greater' →The frequency of PM10 concentrations greater than

*[Answer] Taken.*

8. As the aerosol loading is greatly increased over East Asia since 1980s, the aerosol loading after 2000 is much larger than that in 1980s. Then, are the aerosol changes shown in Figure 10 dominated by aerosol changes after 2000? Maybe you should normalize the PM10 data for each year before compositing the multi-year mean.

*[Answer] Yes, the AOD data of MODIS are available from 2002 to 2012 and $PM_{10}$ concentrations are for the period of 2001-2012.*

*We used deviation standardization to normalize the $PM_{10}$ data ($x_1, x_2, ..., x_n$) for each year. The normalized $PM_{10}$ data ($y_i$) are deduced by:*

$$y_i = \frac{x_i - \min_{1 \le j \le n}\{x_j\}}{\max_{1 \le j \le n}\{x_j\} - \min_{1 \le j \le n}\{x_j\}} \qquad (1)$$

*The normalized data range from 0 to 1. The normalized results are shown in Figure A1-1. It reveals the $PM_{10}$ decreases significantly during the holidays. The frequency of normalized $PM_{10}$ greater than 0.2 during days [-15, -11] is 44.1% while the frequency for the same bin during days [+1, +5] is 23.9%. In Figure A1-1(b), the mean value for days [-4, -1] is -0.023 to eastern China, which is 16% reduction to these four days' climate mean. The normalized results are consistent with the origin and the $PM_{10}$ concentration is more impressive. In the revised version, we added a couple of sentences in Section 4.1 (the last paragraph) to indicate this.*

*In revision, it reads:" Note the $PM_{10}$ composites might be biased by their trends and outliers. To address this question, we repeated the analysis based on yearly normalized data. Here all $PM_{10}$ data are rescaled according to maximum minus minimum range year by year. The normalized data show that the frequency of $PM_{10}$ greater than 0.2 during days [-15, -11] is 44.1% while the frequency for the same bin during days [+1, +5] is 23.9%. The mean value for days [-4, -1] is 16% lower than climate mean. This*

[Figure]

*Figure A1-1. (a) Frequency distributions for normalized PM$_{10}$ concentration during days [-15, -11] (red dashed line) and days [+1, +5] (blue solid line) over southern China. (b) The temporal anomalies of normalized PM$_{10}$ concentration in eastern and southern China. Only years with more than 50% PM$_{10}$ station data available are employed for anomalous composites.*

9. The authors examined the time-lag correlation between PM10 concentration and the anomalous cyclone, and found that the correlation is largest if the PM10 leads by -9 to -6 days. Is it possible that this correlation is due to the 1-2-weeks period of synoptic systems? In other words, the northerlies associated with a synoptic system could decrease the aerosol loading, and it may appear as if the aerosol decrease is correlated with northerlies associated with the next synoptic system that comes in 1-2 weeks. In Figure 11a, the curves rise for positive lead/lag days and may reach a similar height at +10 as that at -9.

*[Answer] Yes, there is 1-2 weeks' variations in natural weather processes which could cause the time-lag correlation between temperature/aerosol concentration and the*

*atmospheric circulation. In our analysis, we think that the signal of natural synoptic processes should have been largely suppressed. Because the natural synoptic system occurrence and their phases of developing are randomly in time. Here we prepared the atmospheric correlation/temperatures time series according to the lunar calendar dates. If there is a natural cyclone around the New Year's Day, its random phase should be offset by other cyclones when put all years together. In other word, no evidence suggests there is a natural cyclone regularly occurs after lunar year's day. We slightly modified the text to mention this in revision (Section 4.2, paragraph 5).*

*In revision, it reads: "It should be pointed out that these time-lag correlations should not be explained by the natural 1-2 week processes. Because the natural synoptic system's occurrence and phases are randomly in time. Here we prepared the atmospheric correlation/temperatures time series according to the lunar calendar dates. If there is a natural cyclone around the New Year's Day, its random phase should be offset by other cyclones when put all years together."*

-THE END-

---

## Author Comment (AC2) · 9 Aug 2018

**General Comments:**

This manuscript presents the anomalous holiday precipitation over southern China during the Chinese Spring Festival based on their analysis of the long-term station observations. The associated meteorological parameters are also analyzed to investigate the possible mechanisms of the reduced precipitation. The manuscript is scientifically sound, well organized, written, and concise. I recommend accepting it as minor revision as below.

**Specific comments:**

P3 L24 it is better to use southern China not China since the results are analyzed in southern China in this study.

*[Answer] Modified.*

P4 L8-9 What is your criterion to exclude the stations? likely if there is only one missing data do you exclude the site?

*[Answer] Yes. Here we exclude the site if there is any missing data.*

P4 L29 what is the step 0?

*[Answer] The ERA-Interim data server surface archive has a mixture of analysis fields, forecast fields and fields available from both the analysis and forecast. If step 0 is chosen, then only analyzed fields, which are produced for 0000, 0600, 1200 and 1800 UTC, are available. ([https://www.ecmwf.int/en/faq/what-are-steps-surface-daily-fields-era-interim](https://www.ecmwf.int/en/faq/what-are-steps-surface-daily-fields-era-interim) )*

P5 L10-18 The statements to calculate the precipitation frequencies are not clear. Actually how many days do you use, 7 days or 3 days? And it contradicts to the 9 days as found with the aerosol time-lag correlations.

*[Answer] Here the calculation of the precipitation frequency anomalies is different from the continuous variables, like temperature. We counted the number of precipitation days (N) of a station in a specific day, like day 0, from 1979 to 2012. Thus, the precipitation frequency (F) is deduced by:*

$$F = \frac{N}{34} \times 100\% \qquad\qquad (1)$$

*To remove the randomness of precipitation, we used a 3-day window. For example, precipitation occurring on the previous day (day -1) and the day after (day +1) is considered to occur on day 0. Above method was also used to calculate the climatic values of F which are based on Gregorian calendar days. For a station from 1$^{st}$ January to 31$^{st}$ March, everyday has a precipitation frequency. Then to this sequence, we used a 7-day window filter to reserve the changes more than 7 days. For a station, in a specific day, like day 0, all precipitation frequencies whose lunar date is day 0 were found from the climatic means sequence and calculated the average. The departure of F from the average is the anomalous frequency ($\Delta$F). Final result is the average of all stations' anomalies. We slightly modified the text to make the statement clearer in revision (Section 2.2, paragraph 3).*

*In revision, it reads: "To remove the randomness of precipitation, we used a 3-day window. For example, precipitation occurring on the previous day (day -1) and the day after (day +1) is considered to occur on day 0. Above method was also used to calculate the climatic values of F which are based on Gregorian calendar days. For a station from 1st January to 31st March, everyday has a precipitation frequency. Then to this climatic sequence, we used a 7-day window filter to reserve the changes more than 7 days because the typical synoptic time scale is approximately 7 days. For a station, in a specific day, like day 0, all precipitation frequencies whose lunar date is day 0 were found from the climatic sequence and calculated the average. The departure of F from the average is the anomalous frequency ($\Delta$F). Final result is the average of all stations' anomalies."*

*We used a 7-day window filter here to remove monthly and sub monthly tendency/variation because February is always warmer than January and second half February is usually warmer than the first half February. Otherwise, the composite result would be biased. Here the rest represents the random synoptic signals within 7 days.*

*The 9-day time-lag correlation was calculated between year-to-year PM$_{10}$ and circulation time series according to the lunar calendar dates. It doesn't mean there is a 9-day cycle. It reveals the correlation is the best when there is a 9-day time-lag*

*between $PM_{10}$ and circulation during holidays. Such correlation between the year-to-year variation of circulation and $PM_{10}$ reflects that year-to-year variation has a fixed time-lag of 9 days, not a relationship on the synoptic scale. We slightly modified the text to avoid misleading for readers in revision (Section 4.2, paragraph 4).*

*In revision, it reads: "It reveals the correlation is the best when there is a 9-day time-lag between $PM_{10}$ and circulation during holidays. Such correlation between the year-to-year variation of circulation and $PM_{10}$ reflects that year-to-year variation has a fixed time-lag of 9 days, not a relationship on the synoptic scale."*

P6 L10-15 What do you mean of the specific day here?

*[Answer] We have changed 'a specific day' to 'each lunar day'. Here we mean the lunar days, like day-1, day0 and day+1, etc.*

P6 L30 It looks there are positive departures.

*[Answer] Yes. It seems to be positive departures of precipitation amount before the Lunar New Year's Day (LNYD). However, they are not evident as they are in the range of 10-90th percentile. At the same time, they are not significant by the Monte-Carlo test. To avoid misunderstanding, we deleted this sentence "Unlike the precipitation frequency, the amount shows no evident departures before the LNYD."*

P7 L16 section→ subsection

*[Answer] Modified. Thanks.*

P7 L29 could you also shown this sub-region in the Figure?

*[Answer] The sub-region (108°E-115°E and 28°N-32°N) is shown in Figure A2-1. The description about the ΔF reduction center in the text is not very appropriate. To avoid misunderstanding, we slightly modified the text to "Most stations located in the north and east of study area have the significant ΔF reduction."*

[Figure]

*Figure A2-1. Anomalies of the precipitation frequency (a) and amount (b). The significances are estimated using a Monte-Carlo approach; stations with circles and dots denote that all days have values significant at the 0.1 and 0.2 levels, respectively. Sub-region (108°E-115°E and 28°N-32°N) is shown as blue dashed box.*

P8 L10 factors-> factor

*[Answer] Done.*

P8 L25 Please give the sample numbers of no rain days

*[Answer] During January to February from 1979 to 2012, for 155 stations, excluding the missing data, there are 152,616 samples for no rain days, 48.9% of that for all days. It can also be concluded from Figure 1a that all the precipitation days of January-February in the study area are about 30 days. In revision we added this information (Section 3.3, paragraph 2).*

P10 L2-3 if total cloud cover shows no evident changes but low cloud covers experiences a significant decrease, does it indicate high cloud covers are increased?

*[Answer] In fact, total cloud cover (TCC) also decreases and is similar to low cloud cover. It seems the reduction of total cloud cover is contributed by the decrease of low cloud*

*cover. As shown in Figure A2-2, middle cloud cover and high cloud cover both show no significant changes in southern China. We modified the text in revision (Section 3.3, paragraph 7).*

*In the revision, we changed the sentence to "The results show that both the total cloud cover and LCC reduces, especially the LCC experiences a significant decrease during the New Year's holiday. It seems the reduction of total cloud cover is contributed by the decrease of low cloud cover."*

[Figure]

*Figure A2-2. (a) Observational daytime total cloud cover anomalies from 1980 to 2012. Daily total cloud cover (b), middle cloud cover (c) and high cloud cover (d) of ERA-Interim from 1979 to 2012.*

Figure 9 is it the horizontal wind or wind anomaly?

*[Answer] Here is the wind anomaly both in Figure 9b and 9c. Modified.*

P16 L1 why do you selected days [-15, -11]? Is it arbitrary?

*[Answer] Yes, it is selected randomly. Traditionally, holidays begin a couple of days before the Lunar New Year. We chose the days [-15, -11] and [+1, +5] for comparison to reveal the aerosol reduction during Chinese Spring Festival. In the Figure A2-3, we selected days [-12, -8] and the holiday aerosol decrease is also evident. We modified the text in revision (Section 4.1, paragraph 3).*

*In revision, it reads: "We randomly selected days [-15, -11] to compute the*

*preholiday period AOD frequency distribution for comparison. We also tested other time period, like days [-12, -8], and the result is robust."*

[Figure]

*Figure A2-3. Frequency distributions for the AOD (a) and PM$_{10}$ concentration (b) during days [-12, -8] (red dashed line) and days [+1, +5] (blue solid line) over southern China.*

P18 L17 The maximum appears for a time-lag of -9, but aerosol lifetime is generally less than one week? Is there any other mechanism?

*[Answer] The 9-day time-lag correlation was calculated between year-to-year PM$_{10}$ and circulation time series according to the lunar calendar dates. It doesn't mean there is a 9-day cycle. It reveals the correlation is the best when there is a 9-day time-lag between PM$_{10}$ and circulation during holidays. The strength of the cyclone every year during the holiday has a 9-day time-lag phase with the PM$_{10}$. Such correlation between the year-to-year variation of circulation and PM$_{10}$ reflects that year-to-year variation has a fixed time-lag of 9 days, not a relationship on the synoptic scale.*

*Aerosol can impact the holiday precipitation/temperature through instant response of the radiation, short-term circulation response, and partly the response of the preceding emissions. It's likely human activity is the only explanation to the holiday precipitation/temperature anomaly.*

*We slightly modified the text to avoid misleading for readers in revision (Section 4.2, paragraph 4).*

*In revision, it reads: "It reveals the correlation is the best when there is a 9-day time-lag between PM$_{10}$ and circulation during holidays. Such correlation between the year-to-year variation of circulation and PM$_{10}$ reflects that year-to-year variation has a fixed time-lag of 9 days, not a relationship on the synoptic scale."*

P20 L19 It is not accurate to use East Asia here.

[Answer] Modified. 'East Asia'-> 'eastern China'

-THE END-

---

## Author Response (AR1)

Answers to Reviewer#1's comments on **pre-discussion version**

*Thanks a lot for your time and comments. Our point-to-point answers are listed below the comments in blue italic.*

Review of the paper entitled "Anomalous holiday precipitation over southern China" by Zhang et al.

This manuscript presents follow-up results of the authors' previous study (Gong et al. 2014, JGR), concerning aerosol reduction and weather changes around the Chinese Spring Festival (CSF). It used extensive station data and the ERA-Interim reanalysis data, and revealed that human activities associated with the holiday could cause a precipitation reduction in days following the CSF. The topic is very interesting and worth pursuing, especially when considering the large impact of the CSF on temperature and precipitation as shown in this manuscript and the JGR paper. The approach for identifying changes of aerosols and precipitation around the CSF is sound to me.

*[Answer] Thanks for your encouragement.*

My main concern is about the relationship between aerosol changes and the precipitation reduction. The authors stated that the aerosol reduction caused by the economic slowdown causes the surface cooling and an anomalous cyclone, which subsequently yields anomalous northerlies to the southern China and reduces the surface precipitation. I don't think the 10% reduction of aerosols can yield a cooling of about 0.8°C to the surface via aerosol-(cloud-)radiation interactions in such a short time period, where feedbacks of the large-scale meteorology are ignorable.

The aerosol reduction and the cooling occur concurrently, but this does not warranty they have cause-effect relationships. The authors stated the reduced downward longwave flux was responsible for the cooling, which I don't agree to. Aerosol effects on the longwave is considered to be small compared with their effects on the shortwave, and the aerosol reduction should increase solar radiation reaching the surface, warming the surface and near-surface air. I think the reduced longwave is caused by the cooling rather than causing the cooling. I suggest the authors either provide more persuading proofs for the cause-effect relationship, or weaken their points on this aspect. By the way, aerosol effects on the shortwave and longwave fluxes can be evaluated using the MERRA2 reanalysis data, which have radiation fluxes over full sky, clear sky and no-aerosol sky separately.

*[Answer] We agree that this is a critical question i.e. what factors have caused such cooling of -0.8ºC and how. The possible roles played by aerosol direct, indirect, and atmospheric feedback all may be important. When considering the simultaneous temperature changes in association with short period aerosol anomaly, the atmospheric feedback would be ignorable. During a moderate period such as >3 days to one week, the atmospheric feedback is likely discernible. This is evident in Figure 13 where the cyclone-temperature correlation peak lags that for PM10*

*a couple of days. In addition, we should point out that the gradual economic slowdown and the emission reduction actually begin long before the New Year's Day, though the largest reduction occurs around the holidays. If this is true, the negative minima of PM10 and temperature during days [-3, -1] would not be totally explained as an instant effect of holiday aerosol. In addition, although aerosol itself is less efficient than moisture and cloud in radiating long-waves, the accompanying northerly, as well as the related clear-sky/less moisture atmosphere/less cloud (Figures 4-9) would help strengthen the temperature anomalies through longwave radiation. In a word, the holiday cooling might be partly due to the instant response of the radiation, partly due to short-term circulation response, and partly due to the response of the preceding emissions. Unfortunately, in our observational analysis we can neither quantitatively confirm any or all of their roles, nor rule out any of them, though our diagnostics and correlations are indicative of their links.*

*As you suggested, we analysed the changes of atmospheric radiative forcing around the holidays by comparing MERRA2 reanalysis productions with/without assimilation aerosols. The top-of-atmosphere (TOA) net radiation anomalies show that significant negative anomalies appear on days [-4, -1] and [+3, +6] in both the MERRA2 products with/without assimilation aerosols (Figure A1). We compared the changes in the net shortwave and longwave radiation, and found that the net radiation anomalies can be largely attributed to the longwave radiations. The means longwave anomalies for these two short periods are respectively -1.77 and -1.57 $Wm^{-2}$ in no aerosol MERRA2 dataset. Meanwhile, the means for these two periods are -1.79 and -1.58 $Wm^{-2}$ in MERRA2 with aerosols. The differences between with and without aerosol are not evident. ERA-Interim has similar results (Figure A1). The analysed radiation datasets in MERRA2 and ERA-Interim are both under clear sky condition. TOA longwave radiation reflects equivalent temperature of air column, smaller TOA upwelling longwave radiation means a cooler beneath atmosphere. The negative TOA longwave radiation and temperature after the New Year's Day would be related to the cooling in association with the anomalous cyclone, whereas the simultaneous influence of atmospheric circulation on the cooling and negative TOA longwave radiation in days [-4, -1] should be small since there is no significant anomalous circulation. Generally, the TOA net radiation and longwave anomalies seem reasonably consistent with the observed temperature changes.*

*Note the aerosol type information in MERRA2 is not available. It seems that the assimilation of AOD ignores the daily aerosol type changes, while the pollutants such as SO2, NOx changes evidently during holidays (e.g., Lin and McElroy 2011; Gong et al., 2014). Without aerosol type information, it is still hard to understand MERRA2 radiation effects by absorbing or by scattering. In the revised version, following the composites of PM10/AOD we have added the analysis of MERRA2 radiation with a short discussion (Section4.1, the last paragraph).*

*Finally, as you suggested, we have weakened our statement in the revised version by modifying the text to simply indicate that there are negative temperature anomalies are found around the holidays in observation, and avoid to directly attribute the $0.8^0C$ cooling to the aerosol radiation. Section 4 has been re-organized. In the revision, we first show there is a significant reduction in air pollution during holidays, and then show that the year-to-year variations of atmospheric circulation after the New Year's Day is statistically correlated with the preceding*

*PM10 concentration with a stable lag time of approximate 6-9 days. Although we avoided to address the cause-and-effect mechanism, these observed correlations and diagnostics should shed lights on their links. This critical issue surely requires further in-depth analysis including modelling in future studies.*

[Figure]

*Figure A1. Top-of-atmosphere (TOA) longwave radiation (a, b) and net radiation (c, d) anomalies around holidays from MERRA2 and ERA-Interim.*

-THE END-

[revised manuscript text omitted]

---

## Referee Report (RR1)

**Comments on the manuscript "Anomalous holiday precipitation over southern China" by Zhang et al. (2018)**

I am glad that the authors accepted my previous advice and revised the manuscript accordingly to some degree. I am fully satisfied with the scientific findings presented. However, I still cannot recommend its publication at this phase. There are two major issues.

First, since the authors no longer attributed precipitation changes to aerosol loading changes, why did they take so much space to talk about aerosol changes during the CSF? The rationales should be given clearly to avoid confusing or misleading readers. Currently, that part seems quite irrelevant to the major topic, especially for the Abstract part.

Second, the authors should spend some time on the English writing in the next version. The manuscript now is too verbose, informal and sometimes confusing or misleading, and has many grammar errors. Below, I list a few examples and hope the authors can check and revise the whole manuscript accordingly.

1. What's the relationship between Chinese Spring Festival (CSF), CSF holiday and Lunar New Year's Day? It could be clear for Chinese readers, but not necessarily for readers from other cultures. The authors should explain it clearly at the beginning of the Abstract, or readers may have difficulties understanding results below.

2. In 'this manuscript reports that during the holidays …..has been significantly reduced', I think 'holidays' should be replaced with 'holiday'; 'significantly reduced' compared with what period? A better statement could be 'the precipitation during the CSF holiday over southern China tends to be lower than days before and after the holiday'.

3. The authors has a mixed use of present and past tenses in describing their results, such as 'the $\Delta$RH showed an evident', 'the $\Delta$RH vertical profiles displays …', 'low cloud cover decreased …', 'LCC also shows a…' They had better stick to one throughout the manuscript.

4. When citing papers, the authors also should pay attention to proper and consistent use of tenses. For example, Page 3, line 3: 'PM2.5 in Beijing-Tianjin-Hebei were reduced' vs. Page 3, line 19: 'there are significant negative …' Both are observational results published in previous studies, why use different tenses?

4. Page 2, line 6-9, the two adjacent sentences both start with 'Recent'. Better use a different transition word.

5. Page 1, line 20 'there are also discernible weekly cycles' in what aspect?

6. Page 2, line 20-25, these sentences are too verbose. They can be shortened as 'there are also discernible weekly cycles in meteorological parameters. For example, Gong et al. (2007) found that there are robust weekly co-variations in the surface solar radiation, total cloud cover,

maximum temperature, and relative humidity in China, where the weekdays tend to have lower total cloud cover and relative humidity but higher surface solar radiation and temperature than the weekends'. Try to condense your words.

7. Page 2, line 26, 'Choi et al. (2008) found that the summertime cloudiness shows a bell-shaped curve between midweek and weekend'. This sentence is confusing. What parameter shows the bell shape in what way? I checked the paper, where the bell-shaped curve referred to interdecadal variations. I could not get this information or even got wrong information from the authors' saying.

8. Page 3, line7: 'for example, on' → such as

9. Page 3, line 10: 'The emission may differ among these events' differ in what aspect? The sentence tells me nothing.

10. Remove all 'etc.' from the manuscript. It is too informal for a scientific paper.

11. Page 4, the last line, 'Lunar New Year's Days (LNYD)', LNYD→LNYDs; replace all 'New Year's Day' afterward with 'LNYD'

12. Page 5, line 14, everyday → every day.

13. Page 5, line 17, 'whose lunar date is' → 'whose lunar dates are'; 'and calculated the average' grammar error

13. Page 16, line 6, this sentence should go into the introduction section.

I could not check errors and verbose sentences throughout the paper and do not ensure no other errors in places between what I list above. The authors should do it and revise the manuscript accordingly.

---

## Author Response (AR2)

**Answers to Referee#1's comments**

*Thanks a lot for your time and comments. Our point-to-point answers are listed below the comments in blue italic.*

Comments on the manuscript "Anomalous holiday precipitation over southern China" by Zhang et al. (2018)

I am glad that the authors accepted my previous advice and revised the manuscript accordingly to some degree. I am fully satisfied with the scientific findings presented. However, I still cannot recommend its publication at this phase. There are two major issues.

First, since the authors no longer attributed precipitation changes to aerosol loading changes, why did they take so much space to talk about aerosol changes during the CSF? The rationales should be given clearly to avoid confusing or misleading readers. Currently, that part seems quite irrelevant to the major topic, especially for the Abstract part.

*[Answer] We completely removed the part of aerosol changes during the CSF. Now the discussion is only about time-lag correlation between the temperature and anomalous cyclone. We just discussed a little about aerosol in the end. Because Chinese Spring Festival (CSF) is a cultural event that is only related to human beings. Many studies show that air pollution reduces significantly during the CSF holiday. The factors which induce the anomalous atmospheric circulation during the CSF holiday need further research and aerosol is the most likely factor affecting atmospheric physics during the CSF holiday.*

Second, the authors should spend some time on the English writing in the next version. The manuscript now is too verbose, informal and sometimes confusing or misleading, and has many grammar errors. Below, I list a few examples and hope the authors can check and revise the whole manuscript accordingly.

*[Answer] Thank you for your careful checking. All the suggestions have been taken. And we checked the whole manuscript and improved the English writing. We also used the academic English editing from American Journal Experts (https://www.aje.com/) and the editorial certificate is behind. In addition, before submitting the paper to Atmospheric Chemistry and Physics, we used the English language editing service from Springer Nature (https://authorservices.springernature.com/).*

1. What's the relationship between Chinese Spring Festival (CSF), CSF holiday and Lunar New Year's Day? It could be clear for Chinese readers, but not necessarily for readers from other cultures. The authors should explain it clearly at the beginning of the Abstract, or readers may have difficulties understanding results below.

*[Answer] Thank you for your suggestion. We modified the Abstract to explain Chinese Spring Festival (CSF), CSF holiday and Lunar New Year's Day.*

*In revision, it reads:" The Chinese Spring Festival (CSF, also known as the Chinese New Year or Lunar New Year) is the most important festival in China. Lunar New Year's Day (LNYD) is the first day of the Lunar New Year. Traditionally, the CSF holiday begins a couple of days before LNYD and ends on lantern day, lasting for approximately 2 weeks"*

2. In 'this manuscript reports that during the holidays …..has been significantly reduced', I think 'holidays' should be replaced with 'holiday'; 'significantly reduced' compared with what period? A better statement could be 'the precipitation during the CSF holiday over southern China tends to be lower than days before and after the holiday'.

*[Answer] Taken. We also checked the whole manuscript and replaced 'holidays' to 'holiday' in the same cases.*

3. The authors has a mixed use of present and past tenses in describing their results, such as 'the ΔRH showed an evident', 'the ΔRH vertical profiles displays …', 'low cloud cover decreased …', 'LCC also shows a…' They had better stick to one throughout the manuscript.

*[Answer] Modified. We revised the manuscript and now only use present tense in describing our results.*

4. When citing papers, the authors also should pay attention to proper and consistent use of tenses. For example, Page 3, line 3: 'PM2.5 in Beijing-Tianjin-Hebei were reduced' vs. Page 3, line 19: 'there are significant negative …' Both are observational results published in previous studies, why use different tenses?

Page 2, line 6-9, the two adjacent sentences both start with 'Recent'. Better use a different transition word.

*[Answer] All taken.*

5. Page 2, line 20 'there are also discernible weekly cycles' in what aspect?

*[Answer] We added 'in meteorological parameters' in this sentence.*

6. Page 2, line 20-25, these sentences are too verbose. They can be shortened as 'there are also discernible weekly cycles in meteorological parameters. For example, Gong et al. (2007) found that there are robust weekly co-variations in the surface solar radiation, total cloud cover, maximum temperature, and relative humidity in China, where the weekdays tend to have lower total cloud cover and relative humidity but higher surface solar radiation and temperature than the weekends'. Try to condense your words.

*[Answer] We have modified these sentences.*

7. Page 2, line 26, 'Choi et al. (2008) found that the summertime cloudiness shows a bell-shaped curve between midweek and weekend'. This sentence is confusing. What parameter shows the bell shape in what way? I checked the paper, where the bell-shaped curve referred to interdecadal variations. I could not get this information or even got wrong information from the authors' saying.

*[Answer] Yes, the bell-shaped curve is the description to interdecadal variations. To avoid confusing, we replaced this sentence with another paper's results.*

*In revision, it reads:" In Korea, Kim et al. (2009) found that there are more cloudiness and less insolation for Wednesday-Thursday and less cloudiness and more insolation for Monday-Tuesday. Furthermore, weekly periodicities are enhanced especially in autumn, more than 2–3 times as great as those of the annual mean with long-term surface measurements of meteorology (1975–2005)."*

*[Answer] We think you may be referring to this sentence 'The CSF is a cultural tradition that is directly related to the daily activity of human beings, particularly economic activities.' in Page 15, line 6.*

*We have moved this sentence to the fourth paragraph of the introduction section.*

I could not check errors and verbose sentences throughout the paper and do not ensure no other errors in places between what I list above. The authors should do it and revise the manuscript accordingly.

*[Answer] Thank you very much and we have checked the whole manuscript.*

**AMERICAN JOURNAL EXPERTS**

**EDITORIAL CERTIFICATE**

This document certifies that the manuscript listed below was edited for proper English language, grammar, punctuation, spelling, and overall style by one or more of the highly qualified native English speaking editors at American Journal Experts.

**Manuscript title:**

Anomalous holiday precipitation over southern China

**Authors:**

Jiahui Zhang, Dao-Yi Gong, Rui Mao, Jing Yang, Ziyin Zhang, Yun Qian

**Date Issued:**

October 9, 2018

**Certificate Verification Key:**

EF64-5711-A073-12CF-172P

[Figure]

This certificate may be verified at www.aje.com/certificate. This document certifies that the manuscript listed above was edited for proper English language, grammar, punctuation, spelling, and overall style by one or more of the highly qualified native English speaking editors at American Journal Experts. Neither the research content nor the authors' intentions were altered in any way during the editing process. Documents receiving this certification should be English-ready for publication; however, the author has the ability to accept or reject our suggestions and changes. To verify the final AJE edited version, please visit our verification page. If you have any questions or concerns about this edited document, please contact American Journal Experts at support@aje.com.

American Journal Experts provides a range of editing, translation and manuscript services for researchers and publishers around the world. Our top-quality PhD editors are all native English speakers from America's top universities. Our editors come from nearly every research field and possess the highest qualifications to edit research manuscripts written by non-native English speakers. For more information about our company, services and partner discounts, please visit www.aje.com.

**SPRINGER NATURE | Author Services**

**Nature Research Editing Service Certification**

This is to certify that the manuscript titled Anomalous holiday precipitation over the southern China was edited for English language usage, grammar, spelling and punctuation by one or more native English-speaking editors at Nature Research Editing Service. The editors focused on correcting improper language and rephrasing awkward sentences, using their scientific training to point out passages that were confusing or vague. Every effort has been made to ensure that neither the research content nor the authors' intentions were altered in any way during the editing process.

Documents receiving this certification should be English-ready for publication; however, please note that the author has the ability to accept or reject our suggestions and changes. To verify the final edited version, please visit our verification page. If you have any questions or concerns over this edited document, please contact Nature Research Editing Service at support@as.springernature.com.

**Manuscript title:**    Anomalous holiday precipitation over the southern China

**Authors:**    Jiahui Zhang, Dao-Yi Gong, Rui Mao, Jing Yang, Ziyin Zhang, Yun Qian

**Key:**    1C72-7131-81E9-AB98-A43P

This certificate may be verified at **secure.authorservices.springernature.com/certificate/verify**.

Nature Research Editing Service is a service from Springer Nature, one of the world's leading research, educational and professional publishers. We have been a reliable provider of high-quality editing since 2008.

Nature Research Editing Service comprises a network of more than 900 language editors with a range of academic backgrounds. All our language editors are native English speakers and must meet strict selection criteria. We require that each editor has completed or is completing a Masters, Ph.D. or M.D. qualification, is affiliated with a top US university or research institute, and has undergone substantial editing training. To ensure we can meet the needs of researchers in a broad range of fields, we continually recruit editors to represent growing and new disciplines.

Uploaded manuscripts are reviewed by an editor with a relevant academic background. Our senior editors also quality-assess each edited manuscript before it is returned to the author to ensure that our high standards are maintained.

[revised manuscript text omitted]

